# A model for organization and regulation of nuclear condensates by gene activity

Halima H. Schede [1,2,7], Pradeep Natarajan[2,7], Arup K. Chakraborty[2,3,4,5] & Krishna Shrinivas [6] ✉

Condensation by phase separation has recently emerged as a mechanism underlying many nuclear compartments essential for cellular functions. Nuclear condensates enrich nucleic acids and proteins, localize to specific genomic regions, and often promote gene expression. How diverse properties of nuclear condensates are shaped by gene organization and activity is poorly understood. Here, we develop a physics-based model to interrogate how spatially-varying transcription activity impacts condensate properties and dynamics. Our model predicts that spatial clustering of active genes can enable precise localization and de novo nucleation of condensates. Strong clustering and high activity results in aspherical condensate morphologies. Condensates can flow towards distant gene clusters and competition between multiple clusters lead to stretched morphologies and activity-dependent repositioning. Overall, our model predicts and recapitulates morphological and dynamical features of diverse nuclear condensates and offers a unified mechanistic framework to study the interplay between non-equilibrium processes, spatially-varying transcription, and multicomponent condensates in cell biology.

The cellular milieu is organized into dozens of membraneless compartments or biomolecular condensates, many of which form through phase separation[1–3]. Condensates concentrate multiple yet specific biomolecules through a network of multivalent and dynamic interactions[3–5]. Further, condensates exhibit a wide variety of physical and material properties and are actively regulated across the cell cycle[1,2,6]. In the crowded cellular environment, distinct condensates are coupled to non-equilibrium processes such as ATP-dependent chemical fluxes and mechanical remodeling that modulate their emergent properties[7,8]. This is particularly evident in the nucleus, where condensates interact with and are regulated by the genome, a large polymeric assembly of proteins, DNA, and RNA. The genome is intrinsically multi-scale and exhibits many layers of organization and regulation - from nanoscale nucleosomal clutches and microloops, to larger micron-scale compartments domains of active and inactive

genes, and nucleus-scale territories for individual chromosomes[9–12]. Across these scales, genome organization is both modulated by and directly modulates active nuclear condensates. Examples include condensates that broadly promote gene expression such as the nucleolus, Histone locus body, nuclear speckles, and transcription-associated condensates, many first observed over a century ago[13–21]. While many studies have emerged in the past decade to probe spatial gene organization[9–12] and condensates[1–3] individually, our understanding of how spatial gene organization and ATP-dependent transcription can affect nuclear condensate behavior and morphology remains nascent. More generally, how this interplay between active processes, multicomponent interactions, and heterogeneous environments dictate condensate properties is poorly understood.

Interactions between RNA and proteins are central in driving the condensation of nuclear bodies[22–25], which in turn promote active

[1]School of Life Sciences, École Polytechnique Fédérale Lausanne, CH-1015 Lausanne, Switzerland. [2]Department of Chemical Engineering, Massachusetts Institute of Technology, Cambridge, MA, USA. [3]Institute for Medical Engineering and Science, Massachusetts Institute of Technology, Cambridge, MA, USA. [4]Department of Physics, Massachusetts Institute of Technology, Cambridge, MA, USA. [5]Department of Chemistry, Massachusetts Institute of Technology, Cambridge, MA 02139, USA. [6]NSF-Simons Center for Mathematical & Statistical Analysis of Biology, Harvard University, Cambridge, MA, USA. [7]These authors contributed equally: Halima H. Schede, Pradeep Natarajan ✉e-mail: krishnashrinivas@g.harvard.edu

transcription at specific genomic loci. Examples include: (1) Histone locus bodies (HLBs) enriched in transcriptional and regulatory proteins as well as multiple genic, enhancer, and small nuclear RNAs that form around the histone gene cluster[26] (2) Transcription-associated condensates which concentrate the transcriptional apparatus as well as noncoding and mRNAs, and preferentially localize to regulatory DNA elements called super-enhancers[27,28] (3) Nuclear speckles which are enriched in the splicing apparatus as well as poly-adenylated mRNAs[29,30] and interfacially localize particular subsets of active genes[29,31,32]. Many nuclear condensates assemble in a manner dependent on active transcription[23,33–35] and exhibit stereotypic localization in the nucleus[24,36–40]. Further, emerging evidence indicates that low and specific levels of non-coding RNA may contribute to the formation of particular genomic or nuclear compartments[24,40]. How specific gene compartments modulate localization or nucleation of condensates through RNA transcription is not well understood.

Unlike simple uniform liquids, nuclear condensates exhibit a wide gamut of transcription-dependent morphologies such as vacuoles[41,42], aspherical shapes[30,39], and layered organization of molecules[43–45] that have been documented over many decades. Yet, how these morphologies arise remain poorly understood. Nuclear condensates also exhibit unusual dynamics including bursts of directed motion[46–49], and inappropriate or aberrant morphologies of nuclear condensates often reflect pathological cell states[39,50,51]. Overall, nuclear condensates are highly variable transcription-dependent compartments with diverse morphologies, dynamics, and localization. Despite their central role in gene regulation, we do not have a unified mechanistic framework to study the emergent properties of nuclear condensates, in large part due to a lack of physically-grounded models of the underlying biology.

In this paper, we build a physically motivated in silico model to explore how actively transcribed gene compartments or clusters modulate condensate properties, and therefore transcription (Fig. 1, Supplementary Fig. 1). We treat gene compartments or clusters in a coarse-grained manner through a spatially varying transcription rate, which reflects the local concentration of actively transcribing genes (Fig. 1b). Through modeling the interplay of spatially varying gene-activity, phase separation, and dynamics of RNA and proteins (Fig. 1c), simulations predict diverse features of active nuclear condensates (Supplementary Fig. 1C). We first identify that compartmentalization or clustering of active genes is sufficient to spatially localize nuclear condensates. At low rates of transcription, we find that clustered genes, through a positive feedback mechanism between local RNA synthesis and resultant protein recruitment, drive de novo nucleation of condensates. At higher rates of transcription from compartmentalized genes, we find that condensates adopt a range of asymmetric and

non-equilibrium steady-state morphologies. When condensates are not proximal to gene compartments, our model indicates that condensates can flow towards distant sites driven by RNA gradients. We subsequently rationalize the limits of this directed motion through simple physical calculations. Finally, we show through simulations that relative clustering, activity, or separation between multiple gene compartments can drive condensates to reposition preferentially to a single compartment, adopt elongated morphologies, or undergo fission. Together, our model provides a unified framework to plausibly explain diverse properties and puzzling observations underlying nuclear condensates (Supplementary Fig. 1C). More generally, our model provides a step towards advancing our understanding of how non-equilibrium processes and gene organization impact regulation and dynamics of multicomponent condensates.

## Results

### Model of active nuclear condensates

Many nuclear condensates enrich molecules that catalyze gene expression[20,25,37] and are proximate to sites of transcription on the genome. The genome itself is organized into spatially clustered hubs of active and inactive genes, referred to as A and B compartments, arising through structural and sequence-based interactions amongst the polymeric DNA scaffold, nuclear proteins, and RNA[9,12,23]. Actively transcribed genes, in turn, modulate both genomic compartments and properties of nuclear condensates in an RNA-dependent manner[11,25,30,52]. How spatial clustering of active RNA synthesis due to structures such as chromatin compartments affects nuclear condensates is poorly understood.

To explore how gene activity affects nuclear condensates, we developed a coarse-grained physics-based model (Fig. 1, Supplementary Fig. 1; Methods) that builds on our previous work[27]. In this model, genomic compartmentalization into active hubs is effectively described by a spatially clustered region of gene activity (Fig. 1b). We model a single active compartment of genes as having a transcription rate constant $k_p(\vec{x})$ that depends on the spatial position $\vec{x}$ (Methods). This spatially varying rate is described by two parameters: the total transcriptional activity ($k_T$) given by the sum of the RNA production rate constant over all spatial positions ($k_T = \int k_p(\vec{x})d\vec{x}$) and the spatial extent of clustering or compartmentalization ($\sigma$). Highly expressed genes contribute to higher $k_T$ while tightly clustered genes correspond to a smaller $\sigma$. Common to many nuclear condensates are high local concentrations of nucleic acids and proteins that often catalyze transcription. To capture this, we employ a coarse-grained description in which protein and RNA components are each modeled as an effective pseudospecies (Fig. 1c, blue and pink species). Attractive interactions

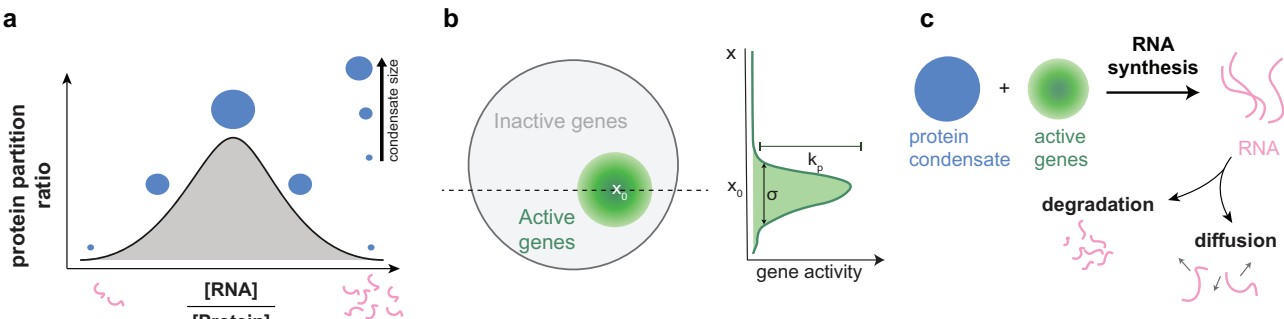

**Fig. 1 | Model of active nuclear condensates.** incorporates the physics of re-entrant phase transitions in RNA-protein systems, spatially clustered gene activity, and active transcription, diffusion, degradation of RNA species to predict condensate behavior. **a** Re-entrant phase behavior of RNA-protein systems: increasing the [RNA]:[Protein] ratio initially increases protein partitioning to the condensate. Once this ratio crosses a threshold set by change balance or entropy-enthalpy

balance, the protein partitioning to the condensate starts to decrease. **b** Spatially varying active genomic regions are represented by a Gaussian distribution centered at $\vec{x}_0$ with spatial extent $\sigma$. The rate constant of RNA synthesis in space is given by $k_p(\vec{x}) = k_p e^{-\frac{\|\vec{x}-\vec{x}_0\|^2}{2\sigma^2}}$. **c** RNA synthesis happens in the condensate, which is defined as a region of high protein concentration. RNA is then degraded at a rate constant $k_d$ and diffuses proportionally to RNA mobility constant $M_r$.

between protein and RNA components promote condensation at particular stoichiometries while preferring the soluble or fully mixed phase at asymmetric stoichiometries (Fig. 1a). This model (Supplementary Fig. 1A) is motivated by an electrostatic complex-coacervate model of transcriptional condensates that we previously developed[27] but extends to condensates whose assembly is primarily driven by heterotypic interactions, as is the case with many biological condensates[53]. The protein and RNA components have specified mobility coefficients and the RNAs are actively synthesized at a rate depending on the density of transcriptional proteins as well as genomic activity and degraded at a uniform rate (Fig. 1c, Methods). The evolution of spatiotemporal dynamics of this system, as described by the RNA and protein concentrations ($\phi_R(x,t), \phi_P(x,t)$), is simulated through evolving a continuum phase-field model (Supplementary Fig. 1B) on a 2D circular grid. Unless specified otherwise, the initial conditions begin with uniform RNA background and a condensate nucleus proximate to the site of genomic activity (see Methods). The simulation data is analyzed (see Methods) to obtain measurements of condensate size, stability, morphology and dynamics. Molecularly quantitative models that capture the complexity of the underlying dynamics, interactions driving phase behavior, and stochastic nature of transcription are challenging to develop and hard to parametrize due to lack of experimental measurements. These approaches are valuable to investigate specific systems that are experimentally well-studied and have these parametrizations readily available. In contrast, we adopt a coarse-grained framework in this study to mechanistically explain diverse phenomena exhibited by condensates. We constrain key parameters of the free energy functional (Supplementary Fig. 2) as well as the dynamic parameters to broadly be in the range of biophysical observations (Supplementary Table 1). For these parameters, such as the RNA degradation rate, the simulation predictions reported in Figs. 2–4 are robust to quantitative perturbations in parameter values.

## Spatial clustering of gene activity dictates condensate size and nucleation

Across a wide range of cell-types and organisms, transcribed genes are spatially clustered in the nucleus[9,52,54] into compartments known as A-type compartments and are often found adjacent to specific nuclear condensates[31,40]. To explore how compartmentalization influences condensate properties, we ran simulations in contrasting scenarios where genes are clustered ($\sigma = 2$) or uniformly distributed (Fig. 2a) by varying the RNA synthesis rate ($k_T$) while holding other parameters constant. We find that increasing gene activity first promotes, and then subsequently dissolves active nuclear condensates (Fig. 2a, b, Supplementary Fig. 3A–C)- consistent with our previous findings for transcriptional condensates[27]. For the same total rate of RNA synthesis $k_T$, we find that clustered genes result in higher local RNA concentrations compared to a spatially uniform gene density (Fig. 2b), which in turn shifts the regime of condensate stability to lower transcription rates (Fig. 2a; black line – clustered, gray line - uniform). Spatial clustering of genes is sufficient to recapitulate this phenomenon in our model, irrespective of the coarse-grained representation of cluster shape (Supplementary Fig. 3B). Reducing density or compartmentalization of active genes i.e. increasing $\sigma$, has a more modest effect on condensate size (Fig. 2c, d, Supplementary Fig. 3C). Physically, reducing the degree of compartmentalization (increasing $\sigma$) leads to lower local concentrations of RNA. Thus, decreasing clustering leads to diffuse RNA concentrations over larger regions leading to larger condensates (Fig. 2d). At very low compartmentalization (very high $\sigma$), condensates stop growing and become smaller. This happens because when RNA concentrations are diffuse, the low concentrations are insufficient to recruit protein apparatus (Fig. 2d).

Together, our model shows that both activity and extent of spatial clustering of active gene clusters modulates size and stability of nuclear condensates. Importantly, genes that are spatially clustered

can promote condensate stability even at low levels of transcription. This may explain why nuclear condensates often form around clustered genomic regions with a wide range of transcriptional activities, such as super-enhancers (transcriptional condensates), rDNA repeats (nucleoli), and histone-gene repeats (HLBs). Consistent with emerging experimental studies, our model provides a mechanistic framework that implicates RNA and compartments of actively transcribed genes as key players in regulating nuclear condensate organization in cis[24,25,54–56].

Nuclear condensates are subject to dynamic control across the cell cycle and often form around specific genomic loci[6,37,57]. To explore whether genomic compartments can drive assembly of nuclear condensates de novo, we ran simulations where no condensates are initially present (Fig. 2e; Methods). Upon increasing the total activity of the genomic compartment, the model predicts de novo condensate nucleation at the site of active transcription under a relatively broad set of parameter regimes (Fig. 2e, Supplementary Fig. 3E–G). Kinetics of nucleation is faster at higher rates of transcription (Supplementary Fig. 3D; right panel). By simulating a range of gene activities and extent of compartmentalization, our model predicts that moderately clustered regions of gene activity are sufficient to drive local nucleation of condensates (Fig. 2f). Through these simulations, our model offers an insight into the conditions required for nucleation: (i) Nucleation requires spatially localized concentrations of RNA and does not occur upon removal of spatial gene clustering (Supplementary Fig. 3D; left panel) (ii) Nucleation does not happen when the mobility of RNA is much larger than the mobility of protein i.e. $M_r/M_p \gg 1$ (Supplementary Fig. 3F) which reflects the pertinent biological parameter regimes (Supplementary Table 1), and (iii) Heterotypic RNA- protein interactions are necessary to drive nucleation (Supplementary Fig. 3H).Together, this model provides a plausible basis to explain diverse observations of transcription-dependent nucleation of condensates, with prominent examples including paraspeckles[33,34,38], nuclear-stress bodies[33,34], and specific nucleolar layers[39].

## Active nuclear condensates exhibit unusual steady-state morphologies

Condensates exhibit diverse morphologies in cells[5,38,39] unlike the symmetric spherical morphology expected from models of simple Newtonian two-phase fluids. While this discrepancy has been partly ascribed to the viscoelastic nature of many condensates[58], how non-equilibrium processes affect shape is not well described. This is particularly relevant in the nucleus where complex morphologies are often lost upon inhibition of active transcription[39,42,59].

To explore how gene expression modulates condensate morphology, we analyzed changes in condensate morphology upon varying the total gene activity $k_T$ and the extent of compartmentalization ($\sigma$). We find that increasing $k_T$ drives vacuole formation in the condensate (Fig. 3a). Vacuoles are regions of low protein but high RNA that form within an otherwise protein-rich condensate (Fig. 3a bottom panel). Vacuoles form when the dense phase rich in both RNA and proteins becomes locally unstable due to high RNA concentrations that arise from the spatial clustering of highly active genes. These vacuoles are lost upon lowering activity (Fig. 3a, top panel) and have dual interfaces (Fig. 3a; Methods) between the inner RNA-rich protein-poor and outer RNA/protein-poor phases - representing a non-equilibrium core-shell morphology. By simulating a range of activities and compartmentalization ($k_T,\sigma$), we find that vacuole formation happens at an intermediate range of these parameters (Fig. 3b, Supplementary Fig. 4A). Physically, at low activity or weak clustering (low $k_T$, high $\sigma$) RNA concentrations are not locally high enough to create vacuoles and very high activity or compartmentalization completely dissolves condensates – thus, vacuoles form at intermediate activity and compartmentalization under simulation conditions. Another prediction of the model is that RNA

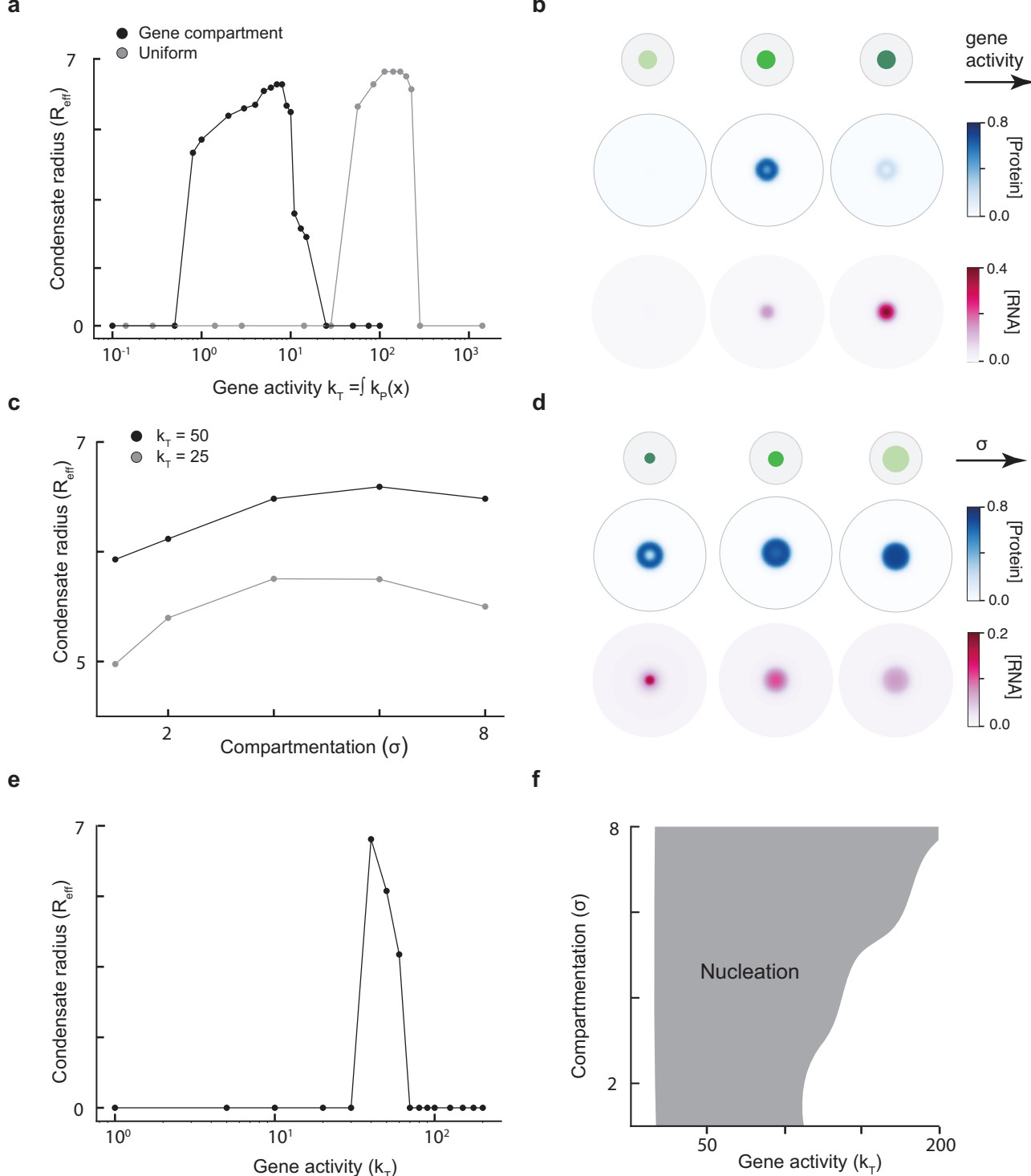

**Fig. 2 | Spatial clustering of gene activity dictates condensate size and nucleation. a** Condensate radius ($R_{eff}$) plotted against gene activity ($k_T$) for two different cases: (i) the gene activity is localized to a region in space with a compartmentalization value of $\sigma = 2$ (black curve) (ii) the gene activity is constant in space (gray curve). For these simulations, a dense phase of protein was nucleated at the center of the simulation domain in a background of dilute protein. **b** Steady state concentration profiles of protein (blue) and RNA (pink) as we increase gene activity ($k_T$) from left to right. **c** Condensate radius ($R_{eff}$) plotted against the extent of compartmentalization ($\sigma$) for two different values of $k_T$. A small value of $\sigma$ corresponds to highly localized gene activity while large $\sigma$ corresponds to uniform gene activity. **d** Steady state concentration profiles of protein (blue) and RNA (pink) as we increase $\sigma$ (decrease compartmentalization) from left to right. **e** Radius ($R_{eff}$) of condensate nucleated by RNA activity alone as a function of gene activity ($k_T$). For these simulations, the initial condition is a uniform dilute protein concentration situated everywhere on the grid. No dense phase of protein was nucleated at the center of the simulation domain. **f** Phase diagram of regions where a condensate is nucleated due to gene activity, upon varying its magnitude ($k_T$) and compartmentalization ($\sigma$). The sharper features of the boundary, for e.g. the kinks, reflect discrete sampling of parameter regimes. Please refer Supplementary Table 2 for details of simulation parameters.

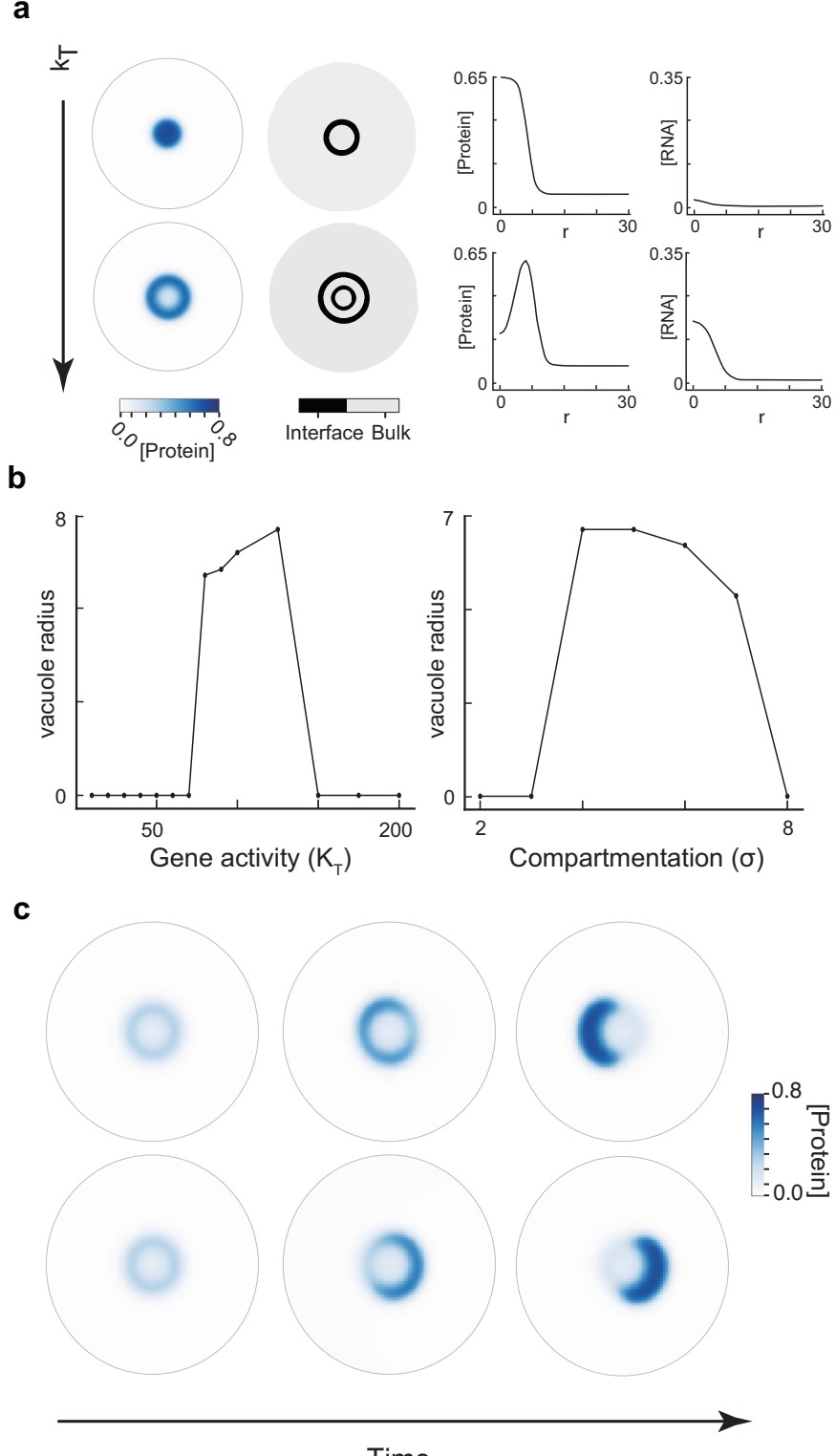

**Fig. 3 | Active nuclear condensates exhibit unusual steady-state morphologies.** **a** In the left panel, the blue plots correspond to the protein concentration profiles and the gray-black plots represent interfaces between two phases for two different values of $k_T$. The right panel shows the concentration profiles of the protein and the RNA as a function of the radial coordinate (r) from the center of the actively transcribing site. In the top panels ($k_T = 1.0$), we have a single dense phase of protein separated from the dilute phase by a single interface. In the bottom panel, increasing the gene activity to $k_T = 20.0$ leads to a ring-like dense phase of protein sandwiched between a dilute phase inside and outside, each separated by respective interface. We call the dilute phase inside the ring as the vacuole. In this way,

increasing gene activity ($k_T$) can lead to a change in condensate morphology from a single droplet (top panel) to a vacuole (bottom panel). **b** Radius of the vacuole plotted against different magnitudes of gene activity $k_T$ (with $\sigma = 4$ fixed) and extents of compartmentalization $\sigma$ (with $k_T = 90.0$ fixed). **c** Protein concentration profiles (blue) as time progresses for two different initial conditions where $k_T$ is large. Both simulations are done for the same values of $k_T$ and $\sigma$. We observe that a symmetric vacuole is initially formed. This symmetry is broken as time progresses and we end up with an aspherical droplet at steady state. Please refer Supplementary Table 3 for details of simulation parameters.

stability modulates vacuole formation (Supplementary Fig. 4F). RNA species that degrade faster (higher $k_d$) drive vacuole formation only at higher rates of transcription ($k_T$).

Nucleoli, condensates built around highly clustered rDNA gene repeats and transcribed at high levels, have been previously documented to contain vacuoles[13,42] and exhibit a multilayered condensate structure, with transcription concentrated within the central regions[39]. Layered morphologies are lost or modified upon inhibition of transcription[39], suggesting an important role for activity. Our model's prediction of vacuoles is broadly consistent with these observations. Further, our model offers an insight into the necessary conditions for vacuole formation: slower diffusion of RNA relative to proteins and an equilibrium re-entrant phase behavior (Supplementary Fig. 4G–I). These conditions are consistent with the pertinent biology - several RNA-protein mixtures exhibit a re-entrant phase diagram driven by heterotypic interactions and most RNA species are often less mobile than proteins, in large part due to their bulkiness and tethering to chromatin (Supplementary Table 1). While prevailing models emphasize the importance of biomolecular interactions[39,60,61] and dynamic processes beside transcription[60,62], our study implicates transcriptional activity as an additional axis regulating nucleolar organization.

At high transcription rates, we find that vacuoles spontaneously break symmetry to form aspherical condensates that partially overlap the site of active transcription (Fig. 3c). Physically, larger activities increase vacuole size, which in turn, increases surface costs due to dual interfaces. This eventually leads condensates to adopt aspherical morphologies with lower surface areas to minimize surface tension costs, even transiently forming 2 aspherical condensates under certain parameter conditions (Supplementary Fig. 4C). Condensates prefer to partially overlap the interface of the active site rather than break into smaller spherical droplets. Physically, this is because spatially decreasing activity and RNA concentration leads to favorable interactions at the interface of the gene that stabilize condensation but surface tension costs limit stability of spherically symmetric vacuoles – leading to adoption of aspherical morphologies. Consistent with this logic, increasing the surface tension at fixed activity in our simulations leads to vacuoles breaking symmetry before eventually dissolving due to high interface costs and vice-versa (Supplementary Fig. 4D). Condensates break symmetry at different positions that are uniformly distributed around the gene across multiple trajectories with slightly different initial conditions, indicating no preferred direction (Supplementary Fig. 4B). Since vacuole formation and aspherical morphologies depend on high local RNA concentrations, we reasoned that increasing the mobility of RNA while holding parameters constant should lower local RNA concentrations. Note that such a perturbation is purely dynamic in nature and does not affect equilibrium properties. We find that lower effective concentrations (or higher $M_r$) leads to a transition from dissolved condensates to aspherical morphologies and eventually vacuoles (Supplementary Fig. 4E), highlighting the importance of dynamics and transport in driving these non-equilibrium morphologies. Overall, our simulation results suggest that strong compartmentalization and high transcriptional activity gives rise to non-equilibrium morphologies like vacuoles and aspherical droplets. This may underlie why nuclear speckles, which are condensates that experience a high RNA flux, often adopt *granular* morphologies which subsequently become spherical upon inhibition of gene expression[59,63] (or loss in RNA flux). Aspherical condensates localized to the edge of active sites are observed in our models, mimicking the interfacial localization of actively transcribed genic regions around nuclear speckles[31,64]. Additional factors likely play important roles to drive in vivo organization and morphology, including interactions between transcriptional and splicing proteins as well as post-translational modifications[64,65], RNA-

dependent changes in interfacial tension, and forces from the chromatin network[66].

## Distant gene activity induces flow of nuclear condensates

Condensates are present in crowded and heterogeneous environments in the cell and typically move randomly, often sub-diffusively[46–48,66], limited by the chromatin or cytoskeletal network. However, many condensates also exhibit bursts of super-diffusive or directed motion[47,48], particularly upon signaling or stress[49]. A mechanistic basis for this directed motion remains lacking.

To explore long-range condensate motion, we developed simulations where the site of gene activity and condensate are initially separated by a distance $r$. Note that our model neglects thermal diffusion of condensates, which is often slow[66] in cells and occurs over minutes-long time-scales. Upon simulating the resultant dynamics, we find that condensates that are initially located far away from the active site flow towards it and upregulate transcription (Fig. 4a, b).

This led us to next ask whether RNA concentration gradients from active sites (generated by basal levels of transcription) contribute to long-range condensate motion - driven by favorable RNA-protein interactions. Consistent with this hypothesis, decreasing the strength of RNA-protein interactions ($\chi$) led to a concomitant decrease in condensate flow velocity, eventually arresting flow in the absence of interactions (Supplementary Fig. 5E). We then sought to relate RNA gradients to the velocity of flow by deriving an analytical estimate of the strength and length-scale of the RNA gradient (Methods 'Theory for flow'), which broadly recapitulated variations in flow velocity with r (Fig. 4c). Our theory predicts that RNA concentrations become smaller far away from the site of gene activity due to turnover and diffusion in the absence of synthesis. As a result, when the distance between the active gene and condensate becomes large, we find that the RNA gradients become too weak to drive flow (Fig. 4c, Supplementary Fig. 5B, Methods 'Theory for flow').

If gradients were driving condensate motion, we hypothesized that perturbations that weaken RNA gradients would result in slower flows. Consistent with this, we find that increasing RNA mobility or increasing RNA degradation in our simulations, both causing weaker gradients through distinct mechanisms, leads to slower condensate motion (Fig. 4d, e). Varying surface tension of the condensate, however, does not alter RNA gradients and therefore does not affect flow velocities (Supplementary Fig. 5C). Rather, lower surface tension led to larger variations in condensate morphology arising from lower energetic costs of deforming interfaces (Supplementary Fig. 5D). Together, our model predicts that if a site of active transcription generates a gradient strong enough to be "sensed" by the condensate, this leads to directed flow.

We next explored whether RNA gradients may be significant enough to drive condensate flow in cells. Based on typical rates of RNA diffusion and transcription in cells (see Methods 'Gradient calculation'), which can vary significantly[67–69], we estimate that gradients in RNA fluxes potentially span a range of scales from about $0.1 - 1\mu m$. These length-scales are comparable to condensate sizes[20], and thus, experimentally discernible if driving short bursts of directed motion. Further, we find that the dimensionless flow velocity predicted by our model (SI Dimensionless flow velocity) in Fig. 4b corresponds approximately to an intracellular velocity (Supplementary Table 1) of $\approx 0.75\mu m/min$. This value falls well within the range of experimentally measured flow velocities of nuclear speckles[49]. While our model neglects thermal diffusion, active nuclear condensates that are otherwise diffusing randomly can exhibit directional motion when a nearby gene compartment becomes active. This provides a mechanistic framework to explain why different nuclear bodies exhibit super-diffusive or partly ballistic motion in their cellular trajectories[46–48] and may provide a mechanism for how nuclear bodies quickly and dynamically allocate machinery upon signaling or stress.

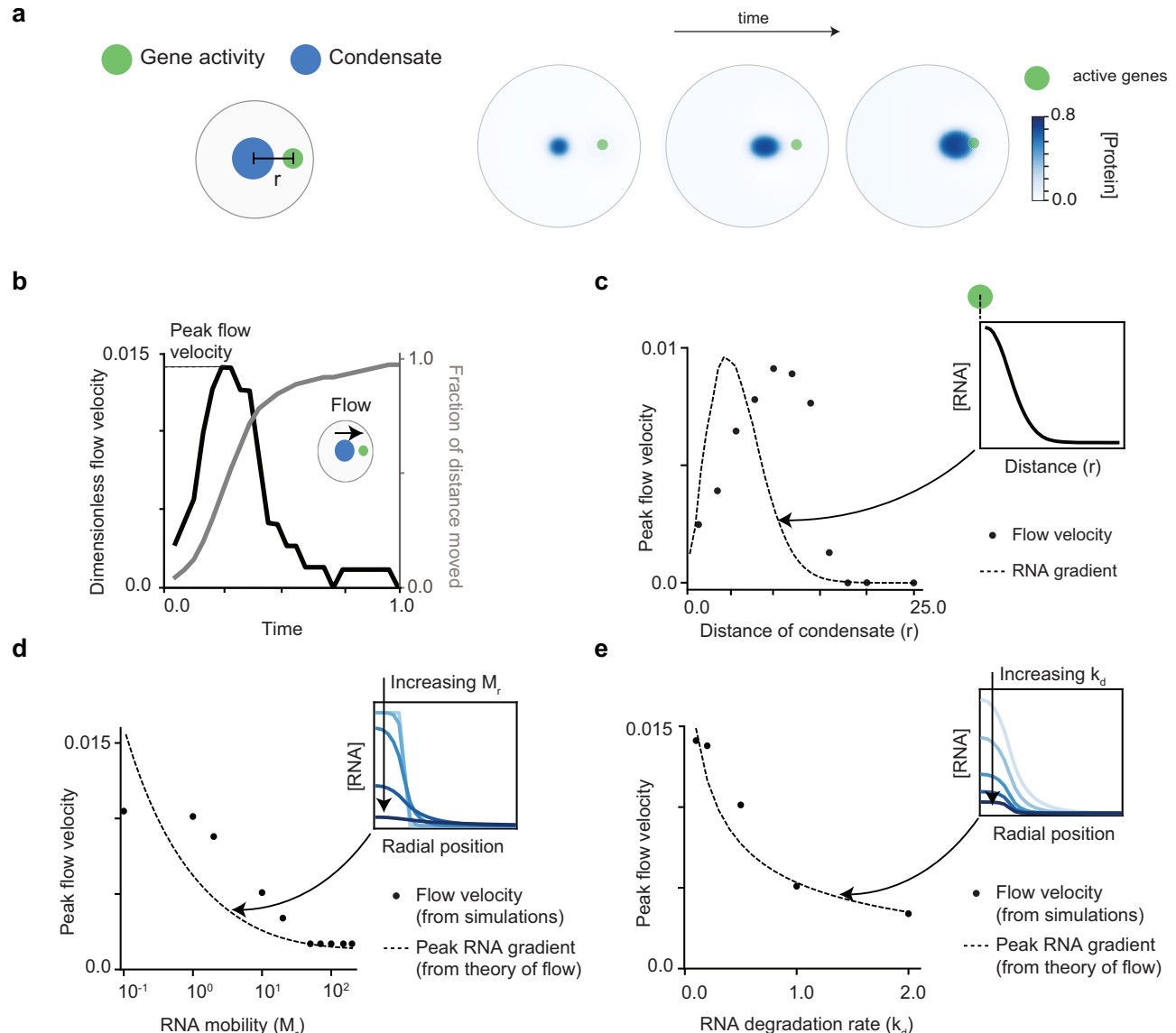

**Fig. 4 | Distant gene activity induces flow of nuclear condensates. a** Cartoon illustrating the initial condition where the center of the condensate (blue) is located at a distance (r) from the center of the site of gene activity (green). The right panel illustrates a distal condensate (blue) moving towards a site of active genes (green) as time progresses. We call this phenomenon as flow. For these simulations, the parameters used are r = 10, $M_r$ = 1.0, and $k_d$ = 0.5. **b** The dynamics of dimensionless flow velocity and fraction of distance moved by the condensate (calculated by displacement from its initial position) over time. Values approximately correspond to experimentally measured intracellular velocity of ≈ 0.75 μm/min seen in nuclear speckles (refer to Supplementary Methods)[49]. **c** Peak flow velocity as a function of distance (r). The flow velocity is highest for an intermediate range of r where the RNA gradient is the highest. **d** Peak flow velocity as a function of RNA mobility ($M_r$). Increasing color intensity corresponds to increasing RNA mobility in the RNA concentration profile plots in the inset. The flow velocity monotonically decreases with $M_r$ as the RNA gradients become weaker due to faster RNA diffusion. **e** Peak flow velocity as a function of the RNA degradation rate ($k_d$). Increasing color intensity corresponds to increasing RNA degradation rate in the RNA concentration profile plots in the inset. The flow velocity monotonically decreases with $k_d$ as the RNA concentrations become smaller due to faster RNA degradation. Please refer Supplementary Table 4 for details of simulation parameters.

## Gene activity and position dictate emergent condensate morphology and dynamics

Given the diverse morphology and dynamics we observed, we sought to derive a phase space of possible outcomes with varying compartment activity and relative initial position of condensate (Fig. 5). We found that condensates were confined to the gene compartment when initially close by (Fig. 5, top panel) irrespective of transcriptional activity. Condensates responded to the presence of a distant active gene compartment by dissolving and subsequently nucleating at the gene compartment, but only when the transcriptional activity was high enough (Fig. 5, top panel). At intermediate distances, condensate dynamics was governed by directed flow (Fig. 5, top panel). These data suggest that dynamics of condensates vary, undergoing directed

motion when close by but not adjacent to a gene compartment, and leading to activity-dependent long-range dissolution and nucleation when present far away. Physically, when RNA concentrations are diffuse, condensates that partially overlap flow up these gradients. However, when condensates are sufficiently far away from a highly active gene cluster, the cluster promotes local nucleation of a condensate. Since overall protein concentrations are fixed in our simulations, motivated by their slow turn-over in cells, this led to a concordant dissolution of the distant condensate. By contrast, the steady-state morphologies of the condensate were nearly independent of initial positions. With increasing gene activity, spherical droplets developed vacuoles, eventually leading to symmetry-breaking (Fig. 3c) and adoption of aspherical morphologies (Fig. 5, bottom panel). At

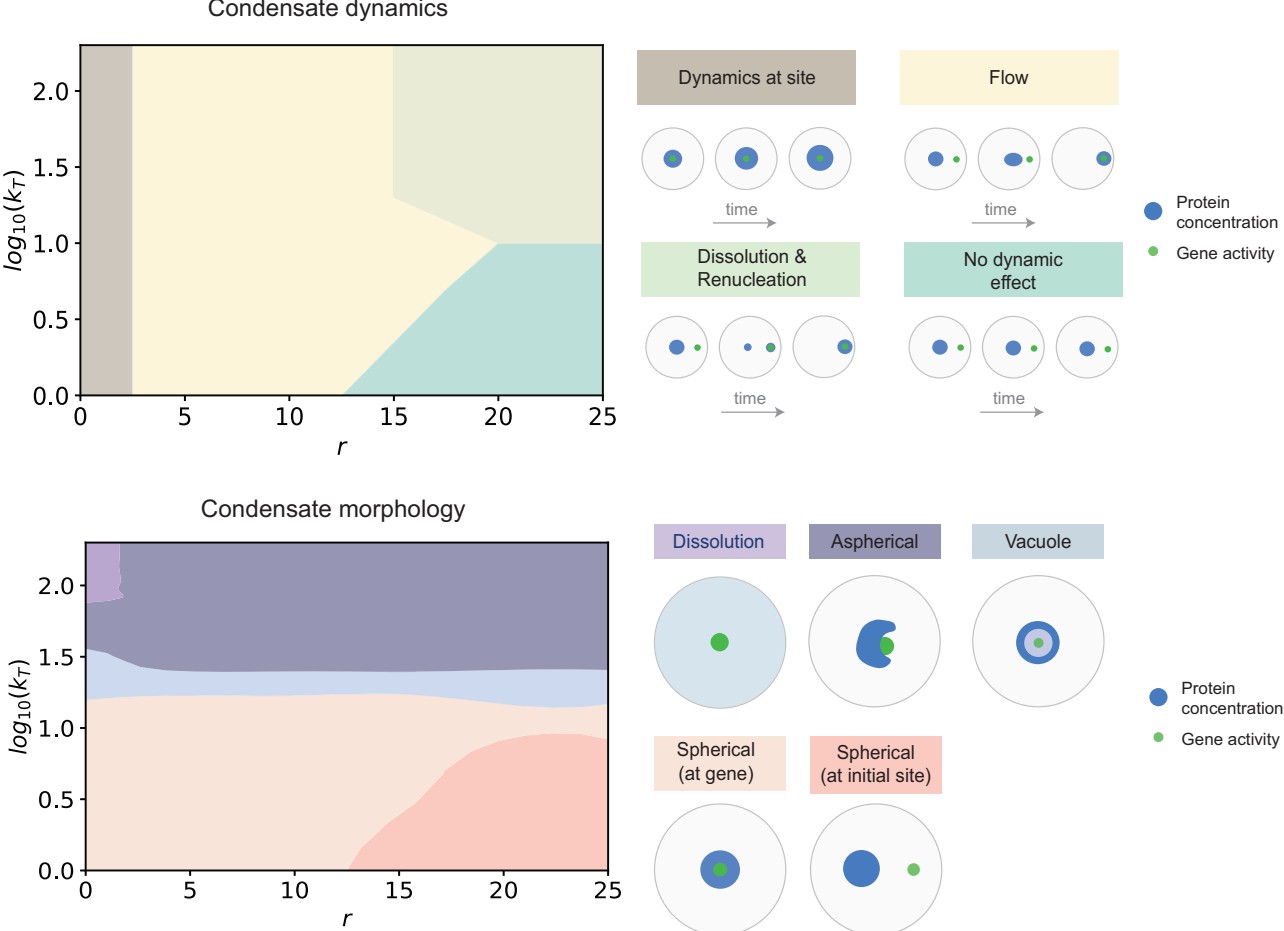

**Fig. 5 | Gene activity and position dictate emergent condensate morphology and dynamics.** The top panel summarizes a diagram of the different qualitative dynamics - dynamics at site, flow, dissolution and renucleation, no effect - that are possible upon varying the distance r and the magnitude of gene activity $k_T$. The bottom panel summarizes a diagram of different qualitative morphologies that are possible upon varying the distance r and the magnitude of gene activity $k_T$: spherical, aspherical, vacuole, dissolution. All simulations were done with $\sigma = 4$. Please refer to Supplementary Table 5 for details of simulation parameters.

very low activities, when the condensates are far from gene clusters, they do not colocalize with clusters as they do not sense the gradient. The primary exception to this lack of dependence on initial positions occurs when condensates are nucleated around high activity genic compartments, leading to their dissolution before repositioning or adoption of different morphologies. Overall, our model partitions condensate dynamics and morphology into two axes - distance from active gene clusters primarily modulates dynamics while transcriptional activity mostly dictates steady-state morphology. This model provides a framework to potentially explain why condensates around regions of high activity, such as nucleoli or nuclear speckles, often exhibit non-equilibrium and aspherical morphologies[33,39,49,59], provides mechanisms by which nuclear bodies exhibit directed or super-diffusive motion[46–48], and shows that compartmentalized gene activity may be sufficient to locally nucleate a condensate[33,34,70].

### Multiple genic compartments compete for or share nuclear condensates

The nucleus of a cell is highly heterogeneous, filled with many different compartments with varying gene activity and multiple condensates[31,32,52] that often interact across large distances. While our primary focus has been to dissect the interplay between nuclear condensates and a single active gene compartment, we next sought to explore how multiple active compartments modulate condensates. To this end, we ran simulations with two active compartments whose properties (activity, extent of compartmentalization, and position) we

changed while holding other parameters constant. Note that we held the total protein levels constant and limiting for all cases. We first considered two nearby compartments - one active (gene cluster B – right compartment in Fig. 6a) and one whose activity we varied (gene cluster A – left compartment in Fig. 6a). In the absence of or at low levels of activity in cluster A, nuclear condensates colocalize with gene cluster B at steady-state (Fig. 6a). As the activity of cluster A is increased, the condensate repositions to overlap both sites (Fig. 6a, ratios = 1, silver line), and when gene cluster A's activity is higher, the condensate adopts an aspherical morphology centered on cluster B but interfacially localizing around cluster A, similar to Fig. 3c. In the latter cases, as activity of gene cluster A is very high, condensates don't completely overlap with the high RNA concentrations. We next explored how changing compartmentalization, but not total activity, of gene cluster A influences condensate morphology. When gene compartments are similar in size but apart by a finite distance $r$, condensates adopt a stretched or elongated morphology to overlap both sites. Decreasing compartmentalization (increasing $\sigma_A$) lessens the local RNA concentrations from cluster A, and so the condensate moves to occupy gene cluster B (Fig. 6b). Conversely, increasing compartmentalization (decreasing $\sigma_A$) increases local RNA concentrations, moving the condensate to cluster A (Fig. 6b). Rather than changing gene features, if we vary the relative position of two genetic clusters, we find that condensates initially elongate to accommodate both sites but eventually splinter to one of two sites (Fig. 6c). This splitting occurs in part due to protein levels being limited. When protein levels are

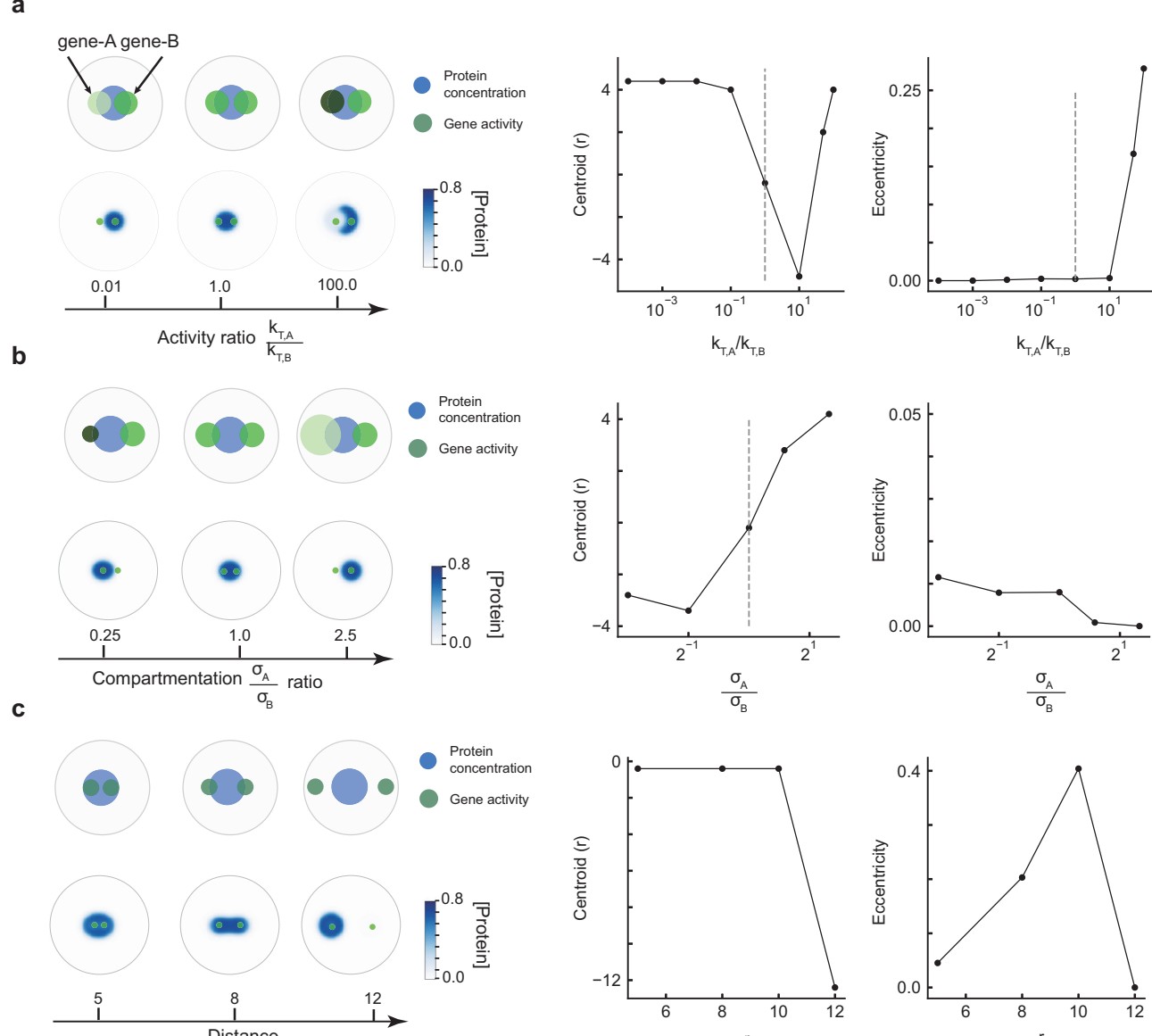

**Fig. 6 | Multiple genic compartments compete for or share nuclear condensates. a** The ratio ($k_{T,A}/k_{T,B}$) of gene activity between gene-A and gene-B is increased by increasing the magnitude of gene activity $k_{T,A}$ from $10^{-4}$ to 100 while keeping $k_{T,B} = 1.0$ fixed. The steady state concentration profile of the protein is shown in blue for different values of the ratio ($k_{T,A}/k_{T,B}$). The right panel plots the centroid and eccentricity of the dense phase of protein for different values of the ratio ($k_{T,A}/k_{T,B}$). Refer the Analyses section of Methods for more details on how the centroid and eccentricity are calculated. **b** The ratio of compartmentation ($\sigma_A/\sigma_B$) of gene activity is increased by increasing $\sigma_A$ from 1.0 to 10.0, while keeping $\sigma_B = 4.0$ fixed. The steady state concentration profile of the protein is shown in blue for different values of the ratio ($\sigma_A/\sigma_B$). The right panel plots the centroid and eccentricity of the dense phase of protein for different values of the ratio ($\sigma_A/\sigma_B$). **c** The steady state concentration profile of the protein is shown in blue for different values of distance ($r$) between the centers of gene-A and gene-B. The right panel plots the centroid and eccentricity of the dense phase of protein for different values of $r$. Please refer to Supplementary Table 6 for details of simulation parameters.

higher, condensates split into 2 droplets rather than elongate when surface tension costs are low (Supplementary Fig. 6). Overall, these simulation results suggest that a combination of compartment strength and activity, as well as their relative positions in the nucleus, contribute to organization and morphology of condensates. When both gene compartments are similar and are active, our model finds that condensates reposition and often elongate to overlap both sites, contributing to aspherical morphologies. This may underlie why RNA Polymerase II condensates, which often span multiple enhancers and genetic loci, exhibit elongated or unfolded shapes in cells[71], as well as other multi-compartment spanning condensates like speckles[59,70]. While the major focus of this section is on condensate morphology and organization, an interesting question lies in exploring how transcription of one gene affects another by modulating condensates. In a

recent study, we explored an aspect of this broad question by investigating how condensate lifetime and transcription of genes are affected by long non-coding RNAs[72].

## Discussion

Condensates are biomolecular assemblies that compartmentalize and organize active cellular processes. In the nucleus, condensates interact with the genome and often regulate active gene expression through concentrating specific molecules or pathways[1,20]. The genome, in turn, is highly organized in the cell, most prominently into territories of individual chromosomes at large length-scales, and sub-divided into many smaller compartments that are broadly enriched in either active or inactive genes[9,11,12]. It is quickly emerging that the interplay between genome organization and nuclear condensates occurs in an RNA/

transcription-dependent manner[5,25,27] and is important for function. Dissecting this complex coupling has remained challenging, in part due to the lack of model frameworks. In this study, we develop a physics-based model to investigate how gene compartments (or hubs) affect nuclear condensates that form by phase separation and the role of active transcription (RNA synthesis) in modulating emergent properties of condensate dynamics, morphology, and organization.

First, we develop a coarse-grained model of the underlying process with three main features (1) Description of nuclear condensate phase behavior in terms of RNA/protein interactions and concentrations (2) Spatial model of gene compartments and (3) Dynamical coupling between condensates and gene compartments through direct activation of transcription when proximate (Fig. 1). Through simulating a range of conditions where features of the gene compartment (activity, extent of compartmentalization) are varied, we identify that spatial clustering can stabilize nuclear condensates even at low levels of gene activity (Fig. 2). Across a broad range of activities and compartment sizes, we find that gene clusters directly and specifically nucleate condensate formation (Fig. 2). Overall, these results from our model indicate that activity and compartment strength directly influence condensate size and dynamics. These results provide a mechanistic basis to interpret why distinct nuclear condensates assemble specifically around clustered genomic elements like super-enhancers[21,73,74], histone-gene repeats[26,36,57], or rDNA repeats[39] and explain how multiple condensates may nucleate in a transcription-dependent manner[34,35,70]. Although we focused on condensates that activate transcription in this study, similarly motivated models will contribute to better understanding of the emerging nexus between inactive heterochromatin and RNA-dependent nuclear condensates[35,75–77] as well as interactions between different types of condensates.

We find that the presence of compartmentalized gene activity, albeit at high levels, gives rise to non-equilibrium steady-state morphologies such as vacuoles and aspherical condensates (Fig. 3). These arise due to high RNA concentrations that locally disfavor the demixed phase and are not observed if activity is lowered. Aspherical or layered morphologies have been previously described for multiple nuclear condensates that abut highly transcribed loci and are sensitive to inhibition of transcription[29,39,43]. Our model provides a mechanistic basis to explain how active RNA fluxes can drive emergent condensate morphologies and likely contribute to morphology in vivo. Through simulations, we next find that condensates can "sense" and exhibit directed motion towards distant compartments (Fig. 4), limited by transport constraints that we estimate through a simple physical calculation of the RNA gradient. This highlights a potential mechanism by which condensates dynamically sense and allocate machinery to genes upon signaling or stress and betters our understanding of diverse yet mechanistically unexplained observations of super-diffusive and long-range motion in nuclear bodies[46–49]. We classify both dynamic and steady-state features of active nuclear condensates, with activity being the primary determinant of steady-state morphology and initial position driving dynamics (Fig. 5). While our model focuses on condensate properties at smaller length and time-scales, an important future area of research will be to dissect how condensates directly restructure chromatin structure over longer-time scales and contribute to large-scale nuclear organization[31,66], including through introduction of capillary forces that arise from condensates wetting different biological substrates[78].

Finally, we find that condensates can exhibit activity-dependent repositioning, elongated morphologies, and fission when a second active compartment is introduced in the vicinity of the first (Fig. 6). These predictions provide a plausible explanation for the organization of Pol II clusters and nuclear speckles, condensates that typically span multiple active sites, into elongated unfolded morphologies[59,71] and observations of condensate fission[47]. This competition between

multiple active sites for condensates may also contribute to the recent phenomenon of condensate hijacking by endogenous retroviruses[79].

Motivated by observations and puzzles in nuclear condensate phenomenology, our model provides a unified framework to interrogate how the interplay between genomic compartments, active transcription, and phase separation influence condensate stability, dynamics, and morphology. Recently, we have developed a related model framework to investigate how gene expression is modulated by transcription of proximal non-coding RNAs[72]. These two models, which build on our previous work[27] in distinct directions, suggest that the interplay of RNA-protein interactions and active transcription plays an important role in organizing the nucleus and in regulating gene expression. Emerging techniques that directly engineer and visualize condensates at specific genomic loci[80–82] provide exciting avenues to test and refine existing models. In turn, new theories and models will likely be required to investigate how multicomponent multiphase fluids form and are modulated by non-equilibrium processes[83–86]. How active processes in general, including chemical fluxes and ATP-dependent molecular motors, couple to genome organization and condensate properties in normal and pathological states represents an important frontier of future research.

Finally, we briefly discuss the limitations of our framework. It is important to note that our model is parametrized by a set of variables whose dimensionless ratios are constrained to pertinent biological ranges (Supplementary Table 1). Specifically, these variables capture biophysical constraints of equilibrium re-entrant phase transitions and low RNA/protein diffusivity ratios. Since our model is coarse-grained and ignores molecular details, it cannot make quantitative predictions about specific molecular species and interactions with condensates. Rather, it serves as a simple unifying framework to understand diverse phenomena characteristic of active nuclear condensate and enables connection of key experimentally amenable parameters (gene activity, RNA concentrations, and position/clustering of genes for example) to condensate phenotypes (including nucleation, vacuole formation, directed flow of condensates, condensate positioning and division). Finally, by building on a continuum framework without incorporating stochasticity or explicit consideration of genome conformation dynamics, our model does not explore the effects of low-copy numbers on phase separation and on larger length-scale links to genome architecture. These factors are likely to be important in shaping condensate morphology under certain conditions for e.g., short-lived gene regulation condensates. More generally, exploring how the diverse network of interactions between specific proteins, regulatory DNA elements, and RNA shapes gene regulation, RNA processing, and nuclear organization for distinct condensates will be an exciting area of future research that will require advances in coarse-grained modeling of multicomponent, heterogenous, and active mixtures.

## Methods
Phase-field simulations and subsequent data analyses were performed using custom code written in python and salient aspects of the model are briefly described in Fig. 1. For a detailed description of the model, simulation, and analyses, please refer below.

### Model
In general, active nuclear condensates contain nuclear proteins and RNA species. These species can interact with each other via a plethora of interactions including, but not limited to - interactions mediated by disordered domains[87] specific structured domains such as RBDs[88], and generic electrostatic interactions. These interactions are captured using a free-energy functional described below:

$$F[\phi_P, \phi_R] = \rho_P(\phi_P - \alpha)^2(\phi_P - \beta)^2 - \chi\phi_P\phi_R + c\phi_P^2\phi_R^2 + \rho_R\phi_R^2 + \frac{\kappa}{2}|\phi_P|^2$$

(1)

Here, $\phi_P$ and $\phi_R$ are concentrations of the nuclear proteins and the RNAs respectively. The first term is a double-well potential that captures protein-protein interactions which drive phase separation, the next two terms capture RNA-protein interactions that result in a re-entrant phase diagram, the last two terms capture the RNA-RNA repulsion and the surface energy of the protein droplet respectively. The choice of parameters used capture the observed phenomenology of RNA-protein phase behavior, similar to a prior study[27]. The interface of the condensate is defined as a region where the determinant of the Jacobian matrix (J) of the free energy functional with respect to the concentration fields becomes negative i.e. det(J) < 0. This is used to define the interface of the condensate in Fig. 3a.

The model used in this paper to study the dynamics of active nuclear condensates is similar to the one used in a prior study[27] and described below. The total amount of protein is treated as a conserved quantity in the time scales of interested and the protein concentration field is assumed to undergo Model B dynamics:[89]

$$\frac{\partial \phi_P}{\partial t} = M_p \nabla^2 \left( \frac{\delta F}{\delta \phi_P} \right) \qquad (2)$$

The dynamics of the protein concentration field is coupled to reaction diffusion dynamics of the RNA species:

$$\frac{\partial \phi_R}{\partial t} = M_r \nabla^2 \phi_R + k_P(\vec{x}) \phi_P - k_d \phi_R \qquad (3)$$

The first term on the right-hand sides captures the diffusion of the RNA species in the nucleus and the last term is a first order decay of the RNA species. The second term on the RHS, the RNA production term, captures the spatial clustering of genic hubs through a spatially varying Gaussian rate constant, which can be interpreted to be proportional to the local gene density. The RNA transcription rate is a product of this rate constant with the protein concentration, reflecting the increased rate of transcription that is expected when higher concentrations of catalyzing protein machinery is present.

## Calculating coexistence concentrations from free energy
The free energy of interaction between the RNA and protein species is given by Eq. (1).

Starting from a total RNA concentration of $\phi_R^0$ and protein concentration of $\phi_P^0$ in the solution, the protein and RNA concentrations in the light and the dense phase can be calculated by solving the following equilibrium relations that equate the chemical potentials of the species and osmotic pressure in the two phases.

$$\mu_P \left( \phi_P^{dense}, \phi_R^{dense} \right) = \mu_P(\phi_P^{light}, \phi_R^{light})$$

$$\mu_R \left( \phi_P^{dense}, \phi_R^{dense} \right) = \mu_R(\phi_P^{light}, \phi_R^{light})$$

$$\Pi \left( \phi_P^{dense}, \phi_R^{dense} \right) = \Pi(\phi_P^{light}, \phi_R^{light})$$

Where the chemical potential of the protein species is $\mu_P(\phi_P, \phi_R) = \partial F / \partial \phi_P$, the chemical potential of the RNA species is $\mu_R(\phi_P, \phi_R) = \partial F / \partial \phi_R$ and the osmotic pressure is $\Pi \left( \phi_P, \phi_R \right) = F(\phi_P, \phi_R) - \mu_P(\phi_P, \phi_R) \phi_P - \mu_R(\phi_P, \phi_R) \phi_R$. In addition, the starting concentration of the species $\phi_P^0$ and $\phi_R^0$ constrain the dense and light phase concentrations in the following way:

$$\nu \phi_P^{dense} + (1 - \nu) \phi_P^{light} = \phi_P^0$$

$$\nu \phi_R^{dense} + (1 - \nu) \phi_R^{light} = \phi_R^0$$

The above system of five equations need to be solved to get the five variables $\phi_P^{light}, \phi_R^{light}, \phi_P^{dense}, \phi_R^{dense}$ and $\nu$. Here, $\nu$ is the volume fraction of the dense phase. These equations are solved to get the phase diagrams in Supplementary Fig. 2 for different parameter values of the free energy.

## Simulations
The model partial differential equations were numerically simulated using a custom python code available at https://github.com/npradeep96/RNA_localization_final. The code uses the finite volume solver Fipy, developed by the National Institute of Standards and Technology[90] under the hood. All simulations in this paper were done in a 2D circular domain of radius 30 units, with a circular discrete mesh. The spatially discretized PDEs were solved for each incremental time step using the sweep() function in Fipy, with adaptive time stepping to pick smaller or larger time steps depending on the how quickly or slowly the concentration fields change. A grid size of $\Delta r = 0.2$ and a typical time step size on the scale of $\Delta t \sim 0.5$ worked well for the simulations. Simulations were run for a duration of about 15,000 time steps, which was sufficient for the system to reach a steady state.

The coexistence concentrations for the protein based on the double well potential are $\alpha = 0.1$ and $\beta = 0.7$. Simulations were done by nucleating a dense seed of protein with a concentration $\phi_P = 0.63$ within a background of dilute protein with a concentration $\phi_P = 0.13$ in the nucleation-and-growth regime. The initial concentration of the RNA species is $\phi_R = 0$ everywhere. The no-flux Neumann boundary condition was applied to all species at the domain boundaries.

For simulations varying the width of the gene-dense region RNA-producing region $\sigma$, the integral of the rate constant i.e. $k_T$ was fixed. Since the rate constant is given by the Gaussian expression $k_P(x) = ce^{-||x-x_0||^2/2\sigma^2}$, the constant $c$ is chosen for each value of sigma such that the integral $k_T = \int_{domain} k_P(x).2\pi x.dx = constant$. An approximate value of the constant given $k_T$ and $\sigma$ is $c \approx \frac{k_T}{\int_0^\infty e^{-||x-x_0||^2/2\sigma^2}.2\pi x.dx} = \frac{k_T}{2\pi\sigma^2}$.

## Analyses
To calculate the radii of the vacuoles if Fig. 3, we calculated the radius of the region of dilute protein concentration $\phi_P < (\frac{\alpha+\beta}{2} = 0.4)$ within the dense condensate of protein with ($\phi_P > 0.4$). The centroid of the protein concentration profiles in Fig. 6 was calculated as the center of mass of this concentration profile $\phi_p(x)$. The eccentricity was calculated as the $e = \frac{(I_{xx} - I_{yy})^2 - 4I_{xy}}{(I_{xx} + I_{yy})^2}$, where $I_{xx}$ and $I_{yy}$ are the second moments of the concentration profile along the x and the y directions, and $I_{xy}$ is the cross moment.

## Gradient calculation
RNAs in the nucleus exhibit a range of diffusivities, with typically chromatin-associated mobilities around $\sim 10^{-3.5} \mu m^2/s$ but higher for transported for mRNPs[68,91]. Depending on the type of RNA species expressed, median half-lives range from $\sim 1$ min (for nc and eRNAs) to 50 min (for lnc and mRNAs)[68] and mammalian RNAs are transcribed across a wide range of rates, with some typical ranges for mRNAs between $\sim 0.5-3$ mRNAs/min[69]. Since its not clear a priori whether the transcription or degradation rates of RNA are limiting in vivo, we estimate an approximate range of gradient length-scales as $l_{off} = \sqrt{M_r/k_d} \approx 0.2 - 1 \mu m$ (assuming similar on-rates) and $l_{on} = \sqrt{M_r/k_p} \approx 0.08 - 0.25 \mu m$ (assuming fast and similar koff rates). These length scales are broadly consistent with the observations that transcribing and non-coding RNAs localize within a micron or so around their site of transcription[40,92].

## Dimensionless flow velocity

For Fig. 4 and Supplementary Fig. S5, we report the velocity of condensate flow in units of $D_P/R_c$, where $D_P$ is the diffusivity of protein and $R_c$ is the radius of the condensate. To relate the protein diffusivity to protein mobility $M_p$ and the free energy parameters $\rho_s, \alpha, \beta$, we equate the coefficient of the $\nabla^2 \phi_p$ term in the expression $M_p \nabla^2(\frac{\delta F}{\delta \phi_P})$ to the protein diffusivity $D_P$. Using this approach, we have the relation $D_P = M_p \rho_s(\alpha^2 + \beta^2 + 4\alpha\beta)$. The condensate radius $R_c$ and the flow velocity $v_f$ are just obtained from the numerical simulations. Using these values, the dimensionless velocity is calculated as $= v_f R_c/D_p$. For the calculations in Fig. 4 and Supplementary Fig. S5, we use a condensate radius of $R_c = 4$ and $D_p = M_p \rho_s \left( \alpha^2 + \beta^2 + 4\alpha\beta \right) = M_p \times 1 \times \left( 0.1^2 + 0.7^2 + 4 \cdot 0.07 \right) = 1.56 M_p$.

## Theory for flow

To derive analytical expressions for the RNA concentration profile and gradient, we set up a transport model with the following assumptions. This approximates the RNA concentration around the site of gene activity when the condensate is far away from it (which is the case at initial times):

- We assume radial symmetry for the problem and solve the equations in a circular domain. There is symmetry about the center of the domain and no-flux boundary condition for the RNA at the edge of the domain.
- The protein concentration at the site of gene activity is assumed to be a constant value of $\phi_P^0$
- The RNA transcription rate is assumed to be a constant value of $k_P$ within a circle of radius $\sigma$ and zero everywhere else to reflect the localized RNA production

Model equations describing localized RNA production, diffusion and degradation in 2D radial coordinates:

$$\frac{M_r}{r}\frac{d}{dr}(r\phi_R) + k_P(r)\phi_P^0 - k_d\phi_R = 0$$

Where:

$$k_P(r) = k_P \; when \; r < \sigma; \; 0 \; otherwise$$

With the boundary conditions:

$$\frac{d\phi_R}{dr}(r=0) = 0$$

$$\phi_R(r \to \infty) = 0$$

Solving these transport equations, we get the expression for the RNA concentration profile as a combination of the modified Bessel functions $I_0$ and $K_0$

$$\phi_R(r) = \frac{k_P\phi_P^0}{k_d}\left(1 - \frac{K_1(\widetilde{\sigma})I_0(r\sqrt{k_d/M_r})}{I_0(\widetilde{\sigma})K_1(\widetilde{\sigma}) + I_1(\widetilde{\sigma})K_0(\widetilde{\sigma})}\right), when\, r < \sigma$$

$$\phi_R(r) = \frac{k_P\phi_P^0}{k_d}\frac{I_1(\widetilde{\sigma})K_0(r\sqrt{k_d/M_r})}{I_0(\widetilde{\sigma})K_1(\widetilde{\sigma}) + I_1(\widetilde{\sigma})K_0(\widetilde{\sigma})}), when\, r > \sigma$$

Where $\widetilde{\sigma} = \frac{\sigma}{\sqrt{M_r/k_d}}$

The largest value of the RNA gradient happens at $r = \sigma$. Using the above equations, we can derive the expression for the max RNA gradient in the domain as:

$$\nabla\phi_R^{\max} = \phi_R^0/L_{dr}$$

Where,

$$\phi_R^0 = \frac{k_P\phi_P^0}{k_d}\frac{(I_0(\widetilde{\sigma}) - 1)K_1(\widetilde{\sigma}) + I_1(\widetilde{\sigma})K_0(\widetilde{\sigma})}{I_0(\widetilde{\sigma})K_1(\widetilde{\sigma}) + I_1(\widetilde{\sigma})K_0(\widetilde{\sigma})}$$

$$L_{dr} = \sqrt{\frac{M_r}{k_d}}\left(\frac{I_0(\widetilde{\sigma}) - 1}{I_1(\widetilde{\sigma})} + \frac{K_0(\widetilde{\sigma})}{K_1(\widetilde{\sigma})}\right)$$

Using the series expansion of the modified Bessel functions about $\widetilde{\sigma} = 0$, we have the following relations to the first order:

$$I_0(\widetilde{\sigma}) - 1 \sim \widetilde{\sigma}^2$$

$$I_1(\widetilde{\sigma}) \sim \widetilde{\sigma}$$

$$K_0(\widetilde{\sigma}) \sim \log\widetilde{\sigma}$$

$$K_1(\widetilde{\sigma}) \sim 1/\widetilde{\sigma}$$

We get the following scaling behaviors of $\phi_R^0$ and $L_{dr}$ on the different parameters:

$$\phi_R^0 \propto \phi_P^0 \frac{\sigma^2 k_P}{M_r}\log\left(\frac{1}{\sigma}\sqrt{\frac{M_r}{k_d}}\right)\left(1 + \sigma^2\frac{k_d}{M_r}\log\left(\sigma\sqrt{\frac{k_d}{M_r}}\right)\right)$$

And

$$L_{dr} \propto \sigma\log\left(\frac{1}{\sigma}\sqrt{\frac{M_r}{k_d}}\right)$$

And

$$\nabla\phi_R^{\max} = \frac{\phi_R^0}{L_{dr}} \propto \phi_P^0\frac{\sigma k_P}{M_r}\left(1 + \sigma^2\frac{k_d}{M_r}\log\left(\sigma\sqrt{\frac{k_d}{M_r}}\right)\right)$$

From analyzing the above functions:

- $L_{dr}$ is a decreasing function of $k_d$ and an increasing function of $M_r$
- $\phi_R^0$ is a decreasing function of $M_r$ (for large $M_r$) and a decreasing function of $k_d$ (at small $k_d$)
- $\nabla\phi_R^{\max} = \phi_R^0/L_{dr}$ is a decreasing function of both $M_r$ and $k_d$

## Fitting theory to simulation data

We postulate that the peak flow velocity must be proportional to the largest value of the RNA concentration gradient in the domain i.e.

$$peak\, flow\, velocity \propto \nabla\phi_R^{\max} = constant.\nabla\phi_R^{\max}$$

The constant that relates peak flow velocity to the concentration gradient was fit with linear least squares using the data points in Fig. 4d, e computed from numerical simulations. The best fit model is depicted as dotted lines in Fig. 4d, e.

## Reporting summary

Further information on research design is available in the Nature Portfolio Reporting Summary linked to this article.

## Data availability

The authors declare that data supporting the findings of this study are available within the paper and/or can be obtained from the simulation code provided below.

## Code availability

The code for running simulations is available at github https://github.com/npradeep96/RNA_localization_final/tree/V_0.0.1 and Zenodo https://doi.org/10.5281/zenodo.8051542[93].

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

## Acknowledgements
K.S. acknowledges support from the NSF–Simons Center for Mathematical and Statistical Analysis of Biology at Harvard (Award #1764269) and the Harvard Faculty of Arts and Sciences Quantitative Biology Initiative. P.N. and A.K.C acknowledge support from NSF (NSF Award #MCB2044895).

## Author contributions
K.S. and A.K.C conceived the project and this research was initiated while H.H.S and K.S. were at M.I.T. H.H.S., P.N. and K.S. developed the model. H.H.S and P.N ran simulations and analyzed data. K.S. designed figures with the help of P.N. and H.H.S. K.S. wrote a first draft and all authors contributed to editing and revising the manuscript.

## Competing interests
A.K.C. is a consultant (titled Academic Partner) for Flagship Pioneering, serves as consultant and on the Strategic Oversight Board of its affiliated company, Apriori Bio, and is a consultant and SAB member of another affiliated company, Metaphore, previously called FL77. He also owns equity in Dewpoint Therapeutics because he previously served on the SAB of this company. All other authors (HHS, PN, and KS) declare no competing interests.
