## [Peer review file · Nature Communications]

REVIEWER COMMENTS

Reviewer #1 (Remarks to the Author):

In the present manuscript, the author aims to explore the relationship between biomolecular condensation, transcription, and genome organization through physics-based modeling. This is a biophysical problem of great importance and a formidable task which necessitates a multi-scale approach to combine the physical nature of genome architecture with models of phase separation and gene expression.

Unfortunately, after reviewing the manuscript, I found the employed model fundamentally flawed, with the manuscript full of problematic hand-holding assertions and lacking in critical evaluation and review of previous work. As a result, I can not recommend the manuscript to any science journals, let alone Nature Communications. Below explain my decision in greater detail:

Authors claim "a physics-based model to interrogate this interplay between condensation, active transcription, and genome organization."

This formidable task necessitates multi-scale models, which at the very least, should resolve the physical features of genome organization. For instance, this has been done in the following papers:

- Fosado, Yair AG, et al. " Proceedings of the National Academy of Sciences 118.10 (2021): e1905215118.
- PR, Cook, D Marenduzzo Nucl Acid Res, 46, 2018
- R Cortini, G J Fillion, Nat Commu, 9, 2018

Instead, authors employ a rather simplistic continuum field model based on the finite volume technique. This has several major flaws: 1. The model lacks the molecular stochasticity needed for modeling dynamics of transcription factors present in low copy numbers.

2. The model lacks any features that one could associate with the genome spatial structure: e.g., the hierarchical organization of loops and domains, epigenetic patterning, and polymeric compartmentalization.

3. The model is not data-driven; it does not incorporate any Hi-C or imaging experiments, and it is not validated against any experimental data, which has become routine in most contemporary models of genome architecture. See, for instance, papers that couple non-equilibrium kinetic processes with genome architecture derived from Hi-C data:

- Chu, Xiakun, and Jin Wang. " Physical Review Letters 129.6 (2022): 068102.
- Eeftens, Jorine M., et al. Nature communications 12.1 (2021): 1-12.
- E Banigan et al, Biorxiv <https://doi.org/10.1101/2022.04.01.486696>

Finally and predictably, most of the results from numerical solutions are represented via cartoonish graphs which lack any realism for genome architecture and, contrary to claims, do not provide any insight into the stochastic molecular origin of non-equilibrium coupling between gene expression, genome architecture, and transcription factor phase-separation.

Reviewer #2 (Remarks to the Author):

In this work Schede H.H. et al. develop an elegant model to describe several aspects of the spatio-temporal interplay between condensation, active transcription, and genome organization. The simulations provide several physical insights that could possibly explain many biological observations, and will be of high interest to the wider community. The paper is very well written. I have a few general remarks to further improve the presentation of the work and the impact for a more biological readership. The work is of high quality and physically sound. The biological relevance of the simulations can be further strengthened.

- The results of the simulations strongly depend on the selected parameters. The choice of the parameters should be discussed in much more details in the Supplementary Tables, specifying references for each parameter, or the rationale behind the choice in the absence of references;
- Connected to the previous point, in several parts of the text the authors refer to experimental observations that are in agreement with the simulations, in one case referring to their previous work. These comments are crucial to show the importance of the findings. The impact of the work would further increase by expanding this comparison, in a quantitative way whenever possible. It is also suggested to incorporate elements of this comparison between experiments and simulations in the figures.
- The model is not described in the main text but only in the supplementary. Several terms are not defined and difficult to follow from the main text only. E.g. line 97: kt, kp, x; line 167 det(J)
- Minor: line 103: Typo "Figure -" ; line 148: "e"

Reviewer #3 (Remarks to the Author):

Nuclear biomolecular condensates are a rich condensate subclass, with interesting additional factors due to their environment, mostly related to the presence of DNA and active transcription. The authors propose a modelling framework that can qualitatively explain features of nuclear condensates, particularly related to their shape and dynamics. The model itself, to my knowledge, has mostly been proposed previously (Ref. 27, Henninger, Oksuz, Shrinivas et al. 2021, Cell). The present study instead explores the model predictions with respect to condensate dynamics and morphology. The paper is unique in that it makes many predictions, none of which are very surprising, but all of which seem to be broadly consistent with the literature. I see the main value of the paper in providing a guideline to which experimental quantifications would be most useful to test our understanding of nuclear condensate formation and maintenance.

General comments:

- Good practice: The model equations (including layout) are an almost exact copy of Ref 27/Fig 4c. In its present form without explicit mention in the figure caption this constitutes self-plagiarism.
 - I really like the fact that the code is available in a well-structured repository.
 - Everything should be dimensional. The claim is applicability to in vivo, so it's necessary to get dimensional diffusion coefficients, etc., to judge whether this is reasonable. Generally given the number of parameters and the lack of any measurements it's hard to make specific claims.
- Input = Output? Many parameters, can obviously explain many phenomena.
- I cannot comment on how common shape changes in the nucleus upon inhibition of transcription are and whether this could be a side-effect, rather than a direct consequence. This argument is crucial to the paper and hopefully another reviewer will be able to comment on this.
 - Would be good to have a table with realistic rates and references next to the simulation parameters used.
 - The discussion is well done and uses a well-balanced language.
 - I feel the paper would benefit from highlighting a few key results more prominently, e.g. vacuoles and flows and could be made more concise otherwise.
- None of this is super surprising, the key strength would be quantitative comparison to data.

- compartmentalization would be a more common term than compartmentation?
- I 79/80 is way too strong. 81/82 is great!
- How does 2D simulation impact conclusions and in particular time scales? 2D and 3D diffusion are very different, and estimating the right time scales is critical to this study's relevance.
- I 141: what does "in cis" mean?

Abstract and Intro

Well written, but I feel uncomfortable with the strong claims based on molecular biology terms. None of these things are 'shown', since you can only 'show' molecular biology results by doing molecular biology, not theory. E.g, line 17 "We show that spatial clustering of active genes enables" could be "We show that in our model spatial clustering of active genes enables..."

Figure 1:

- not immediately clear what the middle columns (protein conc., RNA conc., Gene activity) means and what the lines/shades are supposed to say (is the outer black line the nuclear membrane or the condensate boundary?)

- k_p depends directly on space, so genome organization is not mobile, correct?
- the equation part of the figure is copied over from reference 27 (fig. 4c), also by the authors, this needs to be indicated!

Figure 2:

The in-figure legend of 2a seems wrong (Gene compartment and Uniform cannot be found in panel) 2c: the overall changes are pretty modest, this would presumably change if system size was larger and larger sigma could be achieved?

2f: why is the boundary so wiggly? Is this a numerical artefact? Boundary conditions?

Figure 4:

2b/c: plotting Ldr/r vs peak velocity would be more useful to see the transition from no to flow regime. Also, is this an actual jump or a continuous increase in flow velocity? If the latter, then the threshold needs to be discussed, since it is somewhat arbitrary.

Is flow coupled to a significant increase in condensate volume? This would be a potential experimental check for this mechanism.

Generally, the supplement fig. S3 seems more informative than main fig. 4, but would probably need a finer grained simulation.

Summary of reviewer response

We thank the reviewers for their helpful feedback and suggestions. Reviewers 2 and 3 note that our work is “... of high quality and physically sound”, that we “develop an elegant model” “... several physical insights that could possibly explain many biological observations”, “The paper is unique in that it makes many predictions ...”, and “of high interest to the wider community”. In their critique to improve the study, reviewers bring up concerns regarding model parameterization, connection and motivation to specific biological parameters and experiments, as well as suggestions to better present figures and text. We have incorporated the reviewers’ feedback and worked to address their concerns in this revised manuscript, and this has greatly improved our paper. A point-by-point description of changes are provided (in blue text) in the reviewer response. The major changes are summarized below:

1. Extensive, and new, simulations, theory, and analyses to strengthen and add rigor to the main findings reported in our paper. (Figures 4C-E, S2, S3E-H, S4F-I, S5A-B, S5E, Reviewer figures)
2. Supplementary data and references that connect model parameters and predictions to biologically relevant parameter regimes (SI Table 1, SI Dimensionless flow velocity, and associated text)
3. Substantial textual and figure changes to further emphasize our findings, improve clarity, and explicitly state model limitations through a dedicated section.
(Figures 1, S1C, Supplementary Table 1, Main text lines 17-18, 80, 82, 85, 102-107, 122-129, 143, 169-174, 197-199, 205-210, 233, 239-241, 250-271, 278-281, 364, 409-425, 689, SI lines 27-29, Discussion lines 368-371, 386-390, Model Limitations section, SI Theory of flow)

Separate from these shared concerns, Reviewer 1 suggests that our model is “fundamentally flawed” predicated on their assumption that the goal of this paper is to predict genome structure. This is incorrect. We do not study genome architecture or long-range effects on the scale of the entire genome. Rather, our paper models how nuclear condensates are organized and regulated by spatially varying gene expression on length scales shorter than the entire genome, which to our knowledge, has not been done before. Further, the references Reviewer 1 provides to support their arguments do not model RNA/protein condensates, re-entrant phase behavior, or active RNA synthesis - features that are central and essential to our study. Also, because we study specific phenomena concerning gene regulation on shorter length scales and not genome architecture, we do not employ multi-scale models. Rather, we use phase-field models which have been demonstrated to describe cellular phenomena that are of interest to us as attested to by comparisons with experiments (Henninger*, Oksuz*, Shrinivas* et al., *Cell* 2021). As Reviewer 2 notes, we “develop an elegant model”. To minimize any potential further confusion, we have revised the text to eliminate any statements that might mislead readers into thinking that we are modeling genome structure. Further, in addressing

other comments, we have strengthened the connection of our coarse-grained model to biological experiments. Overall, we believe that Reviewer 1's critique is centered on comparisons to papers that model distinct phenomena - i.e. genome structure - which is not the focus or scope of this study.

Point-by-point response to reviewers

Reviewer #1 (Remarks to the Author):

In the present manuscript, the author aims to explore the relationship between biomolecular condensation, transcription, and genome organization through physics-based modeling. This is a biophysical problem of great importance and a formidable task which necessitates a multi-scale approach to combine the physical nature of genome architecture with models of phase separation and gene expression.

Unfortunately, after reviewing the manuscript, I found the employed model fundamentally flawed, with the manuscript full of problematic hand-having assertions and lacking in critical evaluation and review of previous work. As a result, I can not recommend the manuscript to any science journals, let alone Nature Communications. Below explain my decision in greater detail:

Authors claim "a physics-based model to interrogate this interplay between condensation, active transcription, and genome organization."

This formidable task necessitates multi-scale models, which at the very least, should resolve the physical features of genome organization. For instance, this has been done in the following papers:

- Fosado, Yair AG, et al. " Proceedings of the National Academy of Sciences 118.10 (2021): e1905215118.
- PR, Cook, D Marenduzzo Nucl Acid Res, 46, 2018
- R Cortini, G J Fillion, Nat Commu, 9, 2018

Instead, authors employ a rather simplistic continuum field model based o the finite volume technique. This has several major flaws:

1. The model lacks the molecular stochasticity needed for modeling dynamics of transcription factors present in low copy numbers.
2. The model lacks any features that one could associate with the genome spatial structure: e.g., the hierarchical organization of loops and domains, epigenetic patterning, and polymeric compartmentalization.
3. The model is not data-driven; it does not incorporate any Hi-C or imaging experiments, and it is not validated against any experimental data, which has become routine in most contemporary models of genome architecture. See, for instance, papers that couple non-equilibrium kinetic processes with genome architecture derived from Hi-C data:
 - Chu, Xiakun, and Jin Wang. " Physical Review Letters 129.6 (2022): 068102.
 - Eeftens, Jorine M., et al. Nature communications 12.1 (2021): 1-12.
 - E Banigan et al, Biorxiv <https://doi.org/10.1101/2022.04.01.486696>
4. Finally and predictably, most of the results from numerical solutions are represented via cartoonish graphs which lack any realism for genome architecture and, contrary to claims, do

not provide any insight into the stochastic molecular origin of non-equilibrium coupling between gene expression, genome architecture, and transcription factor phase-separation.

Reviewer 1 suggests that our model is “fundamentally flawed” predicated on their assumption that the goal of this paper is to predict genome structure. This is incorrect. We do not study genome architecture or long-range effects on the scale of the entire genome (Reviewer 1’s major concerns). Rather, our paper models how nuclear condensates are organized and regulated by spatially varying gene expression on length scales shorter than the entire genome, which to our knowledge, has not been done before. Further, the references Reviewer 1 provides to support their arguments do not model RNA/protein condensates, re-entrant phase behavior, or active RNA synthesis - features that are central and essential to our study. Also, because we study specific phenomena concerning gene regulation on shorter length scales and not genome architecture, we do not employ multi-scale models. Rather, we use phase-field models which have been demonstrated to describe cellular phenomena i.e., nuclear condensate phenomenology that are of interest to us as attested to by comparisons with experiments (Henninger*, Oksuz*, Shrinivas* et al., *Cell* 2021). As Reviewer 2 notes, we “develop an elegant model”.

To minimize any potential further confusion, we have revised the text to eliminate any statements that might mislead readers into thinking that we are modeling genome structure.

While we do not specifically model genome architecture, Reviewer 1 (in their third concern) and other reviewers bring up the need for stronger connection between the model we propose and specific biological parameters/experiments. Towards addressing that, we have (this point is repeated in Reviewer 2.1’s concerns) developed and now provide a combination of extensive new simulations, analyses, and references to specific biological variables that are summarized below.

First, we have performed an extensive literature survey and included a Supplementary Table (Table S1) describing biologically relevant ranges of parameters used in this study. We note that the purpose of our paper is to describe new mechanisms rather than quantitatively recapitulate specific data, which are both incredibly hard to measure precisely and vary widely for complex biological systems. Therefore, we have attempted to constrain the parameters to be in a biologically relevant range (as the reviewer states).

Second, we justify the choices of the free energy parameters used in this study through figure S2. Specifically, we look at the free energy parameters χ and c , as they capture the cross-interactions between the RNA and the protein species which couples their dynamics together. We find that both χ and c have to be positive to give rise to a re-entrant phase diagram of the RNA-protein system, which is consistent with experiments (Henninger et al., 2021). Increasing the magnitude of χ and c leads to a qualitatively similar re-entrant phase diagram with the differences largely being in the extent of

protein partitioning to the dense phase and the RNA:Protein ratios at which the partition ratio starts to decrease. Given these observations, we used a value of $\chi = 1.0$ and $c = 10.0$ for the rest of this study, which is consistent with a re-entrant phase transition observed in experiments and qualitatively recapitulates this phase diagram.

We also investigated the impact of using free energy expressions with parameters that are inconsistent with a re-entrant phase transition: by setting $\chi = 0$ or $c = 0$. For the case with $\chi = 0$, there are no attractive interactions between the RNA and the protein species whatsoever, and the protein partition ratio monotonically decreases with the RNA:protein ratio (Figure S2C). In this artificial scenario, our dynamic model predicts that RNA transcription does not lead to condensate nucleation (Figure S3H), vacuole formation (Figure S4I), or flow (Figure S5E). For the case with $c = 0$, the RNA and protein species are always sticky and there are no electrostatic or entropic penalties that prevent phase separation at large concentrations. In this artificial scenario, our dynamic model predicts that RNA transcription can aid condensate nucleation, but does not lead to any condensate dissolution (Figures S3H) or vacuole formation (Figure S4I). We have summarized the same in the below figure (Rev Fig 1).

Reviewer figure 1. (a) Steady-state condensate radii vs k_T using the original free energy expression and comparison with parameter values for free energy, $c = 0$ and $\chi = 0$, that are inconsistent with a re-entrant phase transition. With $c = 0$, we do not get condensate dissolution upon increasing k_T . With $\chi = 0$, we do not have condensate nucleation as well. (b) Vacuole radii using the original free energy expression and comparison with parameter values for free energy, $c = 0$ and $\chi = 0$, that are inconsistent with a re-entrant phase transition. In both these cases, there is no vacuole formation (c) The peak flow velocity of condensates scales linearly with the RNA-protein attraction strength (χ) and becomes 0 when $\chi=0$, indicating that this is the interaction that drives flow.

Third, we investigate the impact of the different dynamical parameters on condensate nucleation, vacuole formation, and flow. For example, the RNA degradation rate, which is a measure of RNA stability, does not affect the qualitative nature of our results. Increasing the RNA degradation rate results in condensate nucleation and vacuole formation at proportionally larger values of k_T (Figures S3E, S4F) and results in flow with a lower flow velocity (Figure 4E), arising from weaker gradients in RNA concentrations. We have summarized the same in the below figure (Rev Fig 2).

Reviewer figure 2. (a) Steady-state condensate radii vs k_T for various RNA degradation rates (k_d). (b) Vacuole radius as a function of k_T for various rates of RNA degradation (k_d). Increasing the RNA stability by decreasing k_d results in vacuoles formation at lower k_T (c) Peak flow velocity for different RNA degradation rates (k_d). Decreasing RNA stability leads to slower flow.

Additionally, our model predicts that condensate nucleation, vacuole formation, and flow do not occur in a parameter regime where the RNA mobility is much larger than the protein mobility i.e. $M_r/M_p \gg 1$ (Figures S3F, S4G, S5A). This reflects the observed biological constraints that RNA often diffuse much slower than proteins, including in part, due to tethering to chromatin or increased bulk. We have summarized the same below (Rev Fig 3).

Reviewer figure 3. (a) Steady-state condensate radii vs k_T for various RNA:Protein mobility ratios (M_r/M_p). (b) Vacuole radius as a function of k_T for RNA:Protein mobility ratios (M_r/M_p). Increasing the mobility ratio results in vacuoles formation at lower k_T (c) Peak flow velocity for different RNA:Protein mobility ratios (M_r/M_p). Decreasing the mobility ratio leads to slower flow.

Finally, we have added text to reflect that the results of the simulations depend on parameters (Section 1 in Results, Lines 126-129, 167-172, 195-197, 203-208), which in turn, are motivated by biological observations.

Reviewer #2 (Remarks to the Author):

In this work Schede H.H. et al. develop an elegant model to describe several aspects of the spatio-temporal interplay between condensation, active transcription, and genome organization. The simulations provide several physical insights that could possibly explain many biological observations, and will be of high interest to the wider community. The paper is very well written. I have a few general remarks to further improve the presentation of the work and the impact for a more biological readership. The work is of high quality and physically sound. The biological relevance of the simulations can be further strengthened.

We thank the reviewer for their positive assessment of our paper as well as the specific feedback for increasing biological relevance and improving presentation. We have worked to incorporate and address all the reviewer's suggestions through a combination of extensive new simulations, text that discusses connections between simulations and experiment, and a new supplementary table that refers to biological data. We have completely redrawn Figure 1, added new figures to supplementary information (S2, S3E-H, S4F-I, S5A-B, S5E) as well as a SI table connecting to pertinent biological parameters (Table S1), and have made significant changes to the other main and supplementary figures to clearly illustrate the impact of different model parameters, justify the choices of parameters used in the study, and highlight the connection with biology in a quantitative way wherever possible. Addressing the comments of this reviewer has greatly strengthened our paper.

1. The results of the simulations strongly depend on the selected parameters. The choice of the parameters should be discussed in much more details in the Supplementary Tables, specifying references for each parameter, or the rational behind the choice in the absence of references;

We thank the reviewer for bringing up this point and have addressed their concern through a combination of extensive new simulations and analyses.

First, we have performed an extensive literature survey and included a Supplementary Table (Table S1) describing biologically relevant ranges of parameters used in this study. We note that the purpose of our paper is to describe new mechanisms rather than quantitatively recapitulate specific data, which are both incredibly hard to measure precisely and vary widely for complex biological systems. Therefore, we have attempted to constrain the parameters to be in a biologically relevant range (as the reviewer states).

Second, we justify the choices of the free energy parameters used in this study through figure S2. Specifically, we look at the free energy parameters χ and c , as they capture the cross-interactions between the RNA and the protein species which couples their dynamics together. We find that both χ and c have to be positive to give rise to a re-entrant phase diagram of the

RNA-protein system, which is consistent with experiments (Henninger et al., 2021). Increasing the magnitude of χ and c leads to a qualitatively similar re-entrant phase diagram with the differences largely being in the extent of protein partitioning to the dense phase and the RNA:Protein ratios at which the partition ratio starts to decrease. Given these observations, we used a value of $\chi = 1.0$ and $c = 10.0$ for the rest of this study, which is consistent with a re-entrant phase transition observed in experiments and qualitatively recapitulates this phase diagram.

We also investigated the impact of using free energy expressions with parameters that are inconsistent with a re-entrant phase transition: by setting $\chi = 0$ or $c = 0$. For the case with $\chi = 0$, there are no attractive interactions between the RNA and the protein species whatsoever, and the protein partition ratio monotonically decreases with the RNA:protein ratio (Figure S2C). In this artificial scenario, our dynamic model predicts that RNA transcription does not lead to condensate nucleation (Figure S3H), vacuole formation (Figure S4I), or flow (Figure S5E). For the case with $c = 0$, the RNA and protein species are always sticky and there are no electrostatic or entropic penalties that prevent phase separation at large concentrations. In this artificial scenario, our dynamic model predicts that RNA transcription can aid condensate nucleation, but does not lead to any condensate dissolution (Figures S3H) or vacuole formation (Figure S4I). We have summarized the same in the below figure (Rev Fig 1).

Reviewer figure 1. (a) Steady-state condensate radii vs k_T using the original free energy expression and comparison with parameter values for free energy, $c = 0$ and $\chi = 0$, that are inconsistent with a re-entrant phase transition. With $c = 0$, we do not get condensate dissolution upon increasing k_T . With $\chi = 0$, we do not have condensate nucleation as well. (b) Vacuole radii using the original free energy expression and comparison with parameter values for free energy, $c = 0$ and $\chi = 0$, that are inconsistent with a re-entrant phase transition. In both these cases, there is no vacuole formation (c) The peak flow velocity of condensates scales linearly with the RNA-protein attraction strength (χ) and becomes 0 when $\chi=0$, indicating that this is the interaction that drives flow.

Third, we investigate the impact of the different dynamical parameters on condensate nucleation, vacuole formation, and flow. For example, the RNA degradation rate, which is a measure of RNA stability, does not affect the qualitative nature of our results. Increasing the RNA degradation rate results in condensate nucleation and vacuole formation at

proportionally larger values of k_T (Figures S3E, S4F) and results in flow with a lower flow velocity (Figure 4E), arising from weaker gradients in RNA concentrations. We have summarized the same in the below figure (Rev Fig 2).

Reviewer figure 2. (a) Steady-state condensate radii vs k_T for various RNA degradation rates (k_d). (b) Vacuole radius as a function of k_T for various rates of RNA degradation (k_d). Increasing the RNA stability by decreasing k_d results in vacuoles formation at lower k_T (c) Peak flow velocity for different RNA degradation rates (k_d). Decreasing RNA stability leads to slower flow.

Additionally, our model predicts that condensate nucleation, vacuole formation, and flow do not occur in a parameter regime where the RNA mobility is much larger than the protein mobility i.e. $M_r/M_p \gg 1$ (Figures S3F, S4G, S5A). This reflects the observed biological constraints that RNA often diffuse much slower than proteins, including in part, due to tethering to chromatin or increased bulk. We have summarized the same below (Rev Fig 3).

Reviewer figure 3. (a) Steady-state condensate radii vs k_T for various RNA:Protein mobility ratios (M_r/M_p). (b) Vacuole radius as a function of k_T for RNA:Protein mobility ratios (M_r/M_p). Increasing the mobility ratio results in vacuoles formation at lower k_T (c) Peak flow velocity for different RNA:Protein mobility ratios (M_r/M_p). Decreasing the mobility ratio leads to slower flow.

Finally, we have added text to reflect that the results of the simulations depend on parameters (Section 1 in Results, Lines 126-129, 167-172, 195-197, 203-208), which in turn, are motivated by biological observations.

2. Connected to the previous point, in several parts of the text the authors refer to experimental observations that are in agreement with the simulations, in one case referring to their previous work. These comments are crucial to show the importance of the findings. The impact of the work would further increase by expanding this comparison, in a quantitative way whenever possible. It is also suggested to incorporate elements of this comparison between experiments and simulations in the figures.

We thank the reviewer for this remark and have addressed this point in the revised study by incorporating references of the various biological parameters and phenomena for comparison.

First, we have added several references (Table S1) to link experimental observations in a more quantitative manner to our findings from model simulations. The table includes references for the observed relative ratios between RNA and protein concentrations, diffusion rates, RNA production versus degradation rates and regimes for flow and connections to the pertinent model parameter regimes.

Second, we have added text under the section “Distant gene activity induces flow of nuclear condensates” comparing our model predictions of flow velocity with experimentally measured values in experiments with the nuclear speckle [Lines 276-279].

“Further, we find that the dimensionless flow velocity predicted by our model (SI Dimensionless flow velocity) in Figure 4B corresponds approximately to an intracellular velocity (SI Table 1) of $\approx 0.75\mu\text{m}/\text{s}$. This value falls well within the range of experimentally measured flow velocities of nuclear speckles⁴⁹.”

However, it is important to note that our model is not a molecularly quantitative framework, which remains a major open challenge in the field due to the inherent biological complexities (also see Model Limitations and Rev response 2.1). To make this point clear, we have added the following additional text in the model definition section [Lines 122-129]:

“Molecularly quantitative models that reflect the complexity of the underlying dynamics, interactions driving phase behavior, and stochastic nature of transcription are challenging to develop and hard to parametrize due to lack of experimental measurements. These approaches are valuable to study specific systems that are experimentally well-studied and have these parametrizations readily available. In contrast, we adopt a coarse-grained framework in this study that can explain diverse phenomena exhibited by condensates. We constrain key parameters of the free energy functional (Figure S2) as well as the dynamic parameters to broadly be in the range of biophysical observations (Table S1).”

Finally, although we agree that displaying a comparison within the figures of previous experiments and simulation would be helpful, we are currently unable to incorporate this since we don't have copyright access to many of the experimental figures. However, we have explicitly included references to papers and specific figures that can be compared with the current figures. We have added the following text to the caption of figure 4B [Lines 689]:

“The dynamics of dimensionless flow velocity and fraction of distance moved by the condensate (calculated by displacement from initial position) over time. Compare this to figure 2D in Kim et al.”

3. The model is not described in the main text but only in the supplementary. Several terms are not defined and difficult to follow from the main text only. E.g. line 97: k_t , k_p , x ; line 167 $\det(J)$

We thank the reviewer for pointing this out and have addressed the concern by revising the text and modifying the figures to improve clarity.

First, we now discuss key terms and parameters that define our model in the main text in the section Model of active nuclear condensates. Specifically, this includes [Lines 102-107]:

“We model a single active compartment of genes as having a transcription rate constant $k_p(\vec{x})$ that depends on the spatial position \vec{x} (SI Model). This spatially varying rate is described by two parameters: the total transcriptional activity (k_T) given by the sum of the RNA production rate constant over all spatial positions ($k_T = \int k_p(\vec{x})d\vec{x}$) and the extent of spatial clustering or compartmentalization (σ , approximate length-scale of clustering).”

We have also added additional clarification of how the condensate interfaces are defined using the determinant of the Jacobian matrix derived from the free energy in SI Model section [Lines 27-29]:

“The interface of the condensate is defined as a region where the determinant of the Jacobian matrix becomes negative i.e. $\det(J) < 0$. This is used to define the interface of the condensate in figure 3A.”

Second, we have significantly modified Figure 1 to better highlight terms that capture the main aspects and parameters of our model, including a new panel B that better illustrates the meaning of the parameters k_p , k_T , and x_0 by visually demonstrating how they model the space to include inactive and active genomic regions.

Minor: line103: Typo “Figure –“ ; line 148: “e”

Thank you - we have corrected the typos.

Reviewer #3 (Remarks to the Author):

Nuclear biomolecular condensates are a rich condensate subclass, with interesting additional factors due to their environment, mostly related to the presence of DNA and active transcription. The authors propose a modelling framework that can qualitatively explain features of nuclear condensates, particularly related to their shape and dynamics. The model itself, to my knowledge, has mostly been proposed previously (Ref. 27, Henninger, Oksuz, Shrinivas et al. 2021, Cell). The present study instead explores the model predictions with respect to condensate dynamics and morphology. The paper is unique in that it makes many predictions, none of which are very surprising, but all of which seem to be broadly consistent with the literature. I see the main value of the paper in providing a guideline to which experimental quantifications would be most useful to test our understanding of nuclear condensate formation and maintenance.

We thank the reviewer for their feedback and for highlighting the uniqueness of our model. We are glad that the reviewer finds our model's predictions easy to follow, which although not surprising to the reviewer, have never previously been unified under a single physical framework and we thus believe represents a worthwhile contribution (as the reviewer notes). In addition, we agree that a significant value of this paper is to emphasize guidelines to test or refine this model. We have now included connections to previous experiments as well as specific tests of model predictions. Further, by addressing the many thoughtful suggestions the reviewer provides below, we believe we have significantly strengthened our model and connection to biological experiments.

General comments:

1. Good practice: The model equations (including layout) are an almost exact copy of Ref 27/Fig 4c. In its present form without explicit mention in the figure caption this constitutes self-plagiarism.

We thank the reviewer for having pointed this out and have made a change to the caption of figure S1B to reflect the similarities and differences of the model equations used in this paper compared to prior work [Lines 282-284]:

"The free energy and dynamic equations are similar to Henninger et al. (2021), with the key difference being the spatially varying gene activity $k_p(x)$ "

Note that the equations are now found in Fig S1 to improve clarity of presentation.

2. I really like the fact that the code is available in a well-structured repository.

Thank you!

3. Everything should be dimensional. The claim is applicability to *in vivo*, so it's necessary to get dimensional diffusion coefficients, etc., to judge whether this is reasonable. Generally given the number of parameters and the lack of any measurements it's hard to make specific claims.

We thank the reviewer for their comment and believe we have largely addressed their concerns through a combination of revised text and the addition of a supplementary table, summarized below. Overall, we want to emphasize that rather than a molecularly quantitative model, our model is coarse-grained but broadly reflects physical and biological parameter regimes, and thus enables modeling of these complex emergent phenomena. To emphasize this point, we have added the following text under the section Model of active nuclear condensates [Lines 122-129]

“Molecularly quantitative models that reflect the complexity of the underlying dynamics, interactions driving phase behavior, and stochastic nature of transcription are challenging to develop and hard to parametrize due to lack of experimental measurements. These approaches are valuable to study specific systems that are experimentally well-studied and have these parametrizations readily available. In contrast, we adopt a coarse-grained framework in this study that can explain diverse phenomena exhibited by condensates. We constrain key parameters of the free energy functional (Figure S2) as well as the dynamic parameters to broadly be in the range of biophysical observations (Table S1).”

To address the concern regarding applicability *in vivo*, we have included references for phenomena occurring *in vivo* in order to link experimental observations in a more quantitative manner to our findings from model simulations (Supplementary Table 1). The table includes measured protein:RNA concentration ratios measured in condensates, biologically relevant ratios of mobility of RNA and protein species, velocities in the case of flow, as well as a column indicating the model's prediction for the relative values.

Our model makes a quantitative prediction about flow velocities *in vivo*. Our simulations predict a dimensionless flow velocity of around 0.01. Converting this to a dimensional velocity using the approach described in SI Dimensionless flow velocity, we get a flow velocity of around $0.75 \mu\text{m}/\text{min}$. This falls within the range of experimentally measured values in nuclear speckles (Kim et al., *J. Cell Science*, 2019). We have added text to reflect this prediction under the section Distant gene activity induces flow of nuclear condensates [Lines 276-279]:

“Further, we find that the dimensionless flow velocity predicted by our model (SI Dimensionless flow velocity) in Figure 4B corresponds approximately to an intracellular velocity (SI Table 1) of $\approx 0.75 \mu\text{m}/\text{s}$. This value falls well within the range of experimentally measured flow velocities of nuclear speckles⁴⁹.”

Our model also shows that condensate nucleation facilitated by RNA transcription, vacuole formation and condensate flow do not happen in a parameter regime where the RNA mobility is much larger than the protein mobility i.e. $M_r / M_p \gg 1$ (Figures S3F, S4G-H, S5A). This is summarized below (Reviewer Figure 3).

Reviewer figure 3. (a) Steady-state condensate radii vs k_T for various RNA:Protein mobility ratios (M_r/M_p). (b) Vacuole radius as a function of k_T for RNA:Protein mobility ratios (M_r/M_p). Increasing the mobility ratio results in vacuoles formation at lower k_T (c) Peak flow velocity for different RNA:Protein mobility ratios (M_r/M_p). Decreasing the mobility ratio leads to slower flow.

Finally, our model also sheds light on the necessary conditions required for condensate nucleation, vacuole formation and flow. We have added the following lines to the main text to clearly highlight these points [Lines 169-174, 205-207, 256-258, 266-268]:

“Our model offers an insight into the necessary conditions for nucleation: (i) Nucleation requires spatially localized concentrations of RNA and does not occur upon removal of spatial gene clustering (Figure S3D; left panel) (ii) Nucleation does not happen when the mobility of RNA is much larger than the mobility of protein i.e. $M_r/M_p \gg 1$ (Figure S3F) which reflects the pertinent biological parameter regimes (Table S1), and (iii) Heterotypic RNA- protein interactions are necessary to drive nucleation (Figure S3H)”

“Further, our model offers an insight into the necessary conditions for vacuole formation: slower diffusion of RNA relative to proteins and an equilibrium re-entrant phase behavior (Figures S4G-I).”

“Consistent with this hypothesis, decreasing the strength of RNA-protein interactions (χ) led to a concomitant decrease in condensate flow velocity, eventually arresting flow in the absence of interactions (Figure S5E).”

“Consistent with this, we find that increasing RNA mobility or increasing RNA degradation, both causing weaker gradients through distinct mechanisms, leads to slower condensate motion (Figures 4D-E).”

These are specific statements that can be used to design experiments that test our model. For example, we predict that disrupting the RNA-protein attractive interactions should prevent *de novo* nucleation of condensates and flow. Increasing the RNA mobility by having short-length RNA species for example should prevent condensate nucleation and vacuole formation. Faster degradation of the RNA species should cause a reduction in the flow velocity.

4. Input = Output? Many parameters, can obviously explain many phenomena.

We thank the reviewer for this remark. To address the concern, we have performed extensive new simulations and calculations (Figures S2, S3, S4 and S5) to demonstrate that the predicted phenomena are only observed when model parameters are constrained to reflect experimentally measured biological constraints.

While our model indeed has many parameters, these parameters correspond to physically meaningful quantities and therefore need to reflect certain constraints. For example, χ corresponds to the strength of protein-RNA attraction and therefore has to be positive as we know that RNA and protein species attract each other at low concentrations (Henninger et al. 2021). Another parameter, c , corresponds to the RNA-protein repulsion at high concentrations and has to be sufficiently positive to capture a re-entrant phase diagram (Henninger et al. 2021, Banerjee et al. 2017). These constraints are important to recapitulate the re-entrant phase diagram (Figure S2). Similarly, the ratio of RNA and protein mobilities are < 1 .

Subject to these constraints on the parameters, we show that the model can exhibit diverse phenomena. This is very different from a neural network for example, where the parameters are relatively unconstrained by any mechanistic reasoning and lack any physical meaning. To demonstrate this, we have added Figure S2 to show that the re-entrant phase separation is supported under only specific regimes of parameters as well as Figures S3H, S4I, S5E that indicate that the model is no longer capable of predicting emergent phenomena when parameters do not reflect the underlying biology. The same is summarized below (Rev Fig 1):

Reviewer figure 1. (a) Steady-state condensate radii vs k_T using the original free energy expression and comparison with parameter values for free energy, $c = 0$ and $\chi = 0$, that are inconsistent with a re-entrant phase transition. With $c = 0$, we do not get condensate dissolution upon increasing k_T . With $\chi = 0$, we do not have condensate nucleation as well. (b) Vacuole radii using the original free energy expression and comparison with parameter values for free energy, $c = 0$ and $\chi = 0$, that are inconsistent with a re-entrant phase transition. In both these cases, there is no vacuole formation. (c) The peak flow velocity of condensates scales linearly with the RNA-protein attraction strength (χ) and becomes 0 when $\chi=0$, indicating that this is the interaction that drives flow.

Additionally, we also performed simulations to assess the sensitivity of the results to the RNA degradation rate, which is a measure of RNA stability, does not affect the qualitative nature of our results (figures S3E, S4F). Increasing the RNA degradation rate results in condensate nucleation and vacuole formation at proportionally larger values of k_T and results in flow with a lower flow velocity, arising from weaker gradients in RNA concentrations (Figure 4E). The same is summarized in the below figure (Rev Fig 2).

Reviewer figure 2. (a) Steady-state condensate radii vs k_T for various RNA degradation rates (k_d). (b) Vacuole radius as a function of k_T for various rates of RNA degradation (k_d). Increasing the RNA stability by decreasing k_d results in vacuoles formation at lower k_T . (c) Peak flow velocity for different RNA degradation rates (k_d). Decreasing RNA stability leads to slower flow.

- I cannot comment on how common shape changes in the nucleus upon inhibition of transcription are and whether this could be a side-effect, rather than a direct consequence. This argument is crucial to the paper and hopefully another reviewer will be able to comment on this.

We thank the reviewer for this comment and have added citations as well as modifications to the text in the results section to provide instances of condensate shape changes in the nucleus in response to alterations in transcription activity.

From a literature review, multiple studies report this phenotype and confirm the observation of shape changes accompanied by fusion events and increases in condensate volume occurred after transcription inhibition (Kim et al. *J. Cell Sci* 2019, Spector and Lamond. *Cold Spring Harb Persp Biol* 2011). One hypothesis for this behavior in the context of nuclear speckles is that these condensates contain splicing factors. The ability of the speckle to split up allows speckle components to redistribute to different active gene loci during periods of high transcription demand as in viral infection. Another example of shape alterations because of a change in RNA transcription was seen in tobacco callus cells where transcription inhibition resulted in the loss of nucleolar vacuoles (Johnson. *Journal of Cell Biology* 1969).

Our model predicts similar changes in response to transcription inhibition. We predicted that active transcription from multiple sites has the potential to stretch the condensate and cause them to split (Figure 6, Figure S5). Under specific conditions it would hence be plausible that transcription inhibition, or a decrease in the number of sites actively transcribing RNA, can cause the inverse and result in fusion, which would explain the reduction in number, increase in volume and roundness of condensates. Finally, we don't claim this is the sole possible mechanism and discuss other potential mechanisms in the main text [Lines 239-241]:

“Additional factors likely play important roles to drive in vivo organization and morphology, including interactions between transcriptional and splicing proteins as well as post-translational modifications^{64,65}, RNA-dependent changes in interfacial tension, and forces from the chromatin network.”

6. Would be good to have a table with realistic rates and references next to the simulation parameters used.

We thank the reviewer for bringing up this point and have now elaborated and addressed this in multiple ways by including both a Supplementary Table of parameters (Table S1) and rationale for choice of parameters and pertinent references (SI figures S2, S3E-H, S4F-I).

7. The discussion is well done and uses a well-balanced language

Thank you!

8. I feel the paper would benefit from highlighting a few key results more prominently, e.g. vacuoles and flows and could be made more concise otherwise. None of this is super surprising, the key strength would be quantitative comparison to data.

We thank the reviewer for pointing this out. We agree with the reviewer that the results are sensible and hence not surprising. This fact allows us to be more confident about the validity of the model which would otherwise be questionable.

Exact quantitative agreement for systems with multiple interacting and reactive species in the context of condensates remains an open challenge. To highlight this point, we have added additional text in the discussion section to highlight our model limitations [Lines 411-417]:

“Since our model is coarse-grained and ignores molecular details, it cannot make quantitative predictions about specific molecular species and interactions with condensates. Rather, it serves as a simple unifying framework to understand diverse phenomena characteristic of active nuclear condensate and enables connection of key experimentally amenable parameters (gene activity, RNA concentrations, and position/clustering of genes for example) to condensate phenotypes (including nucleation, vacuole formation, directed flow of condensates, condensate positioning and division).”

However, we have tried our best to motivate quantitative connections whenever possible. As mentioned before, we have highlighted the results relating to flow and a semi-quantitative comparison with experiments in the results section under Distant gene activity induces flow of nuclear condensates [Lines 278-281]:

“Further, we find that the dimensionless flow velocity predicted by our model (SI Dimensionless flow velocity) in Figure 4B corresponds approximately to an intracellular velocity (SI Table 1) of $\approx 0.75\mu\text{m}/\text{s}$. This value falls well within the range of experimentally measured flow velocities of nuclear speckles⁴⁹.”

9. compartmentalization would be a more common term than compartmentation?

We thank the reviewer for this suggestion and have made the modifications.

10. I 79/80 is way too strong. 81/82 is great!

We thank the reviewer for the suggestion. We have modified the text in these lines to [Line 85]:

*“Together, our computational model provides a unified framework to **plausibly explain** diverse properties and puzzling observations underlying nuclear condensates”*

11. How does 2D simulation impact conclusions and in particular time scales? 2D and 3D diffusion are very different and estimating the right time scales is critical to this study's relevance.

In short, the qualitative features we present should not depend on dimensionality, as expected for this universality class of phase transitions, as we have also previously shown in Hennigner, Oksuz, Shrinivas et al. Cell 2021.

12. l 141: what does "in cis" mean?

In cis was supposed to refer to spatially proximal effects, which we now clarify using simplified language.

Abstract and Intro

Well written, but I feel uncomfortable with the strong claims based on molecular biology terms. None of these things are 'shown', since you can only 'show' molecular biology results by doing molecular biology, not theory. E.g, line 17 "We show that spatial clustering of active genes enables" could be "We show that in our model spatial clustering of active genes enables..."

We thank the reviewer for their suggestion and have suitably softened the language in the text. We have modified words like "show" or "shown" to "our model predicts" or "our model indicates" or "our simulations show".

Specifically, we have changed the wording in the following lines:

Previous: We show that spatial clustering of active genes enables precise localization and de novo nucleation of condensates

Modified: "**Our model predicts that** spatial clustering of active genes enable precise localization and de novo nucleation of condensates" [17-18]

Previous: ... We show that condensates can flow towards distant sites, driven by RNA gradients, and subsequently ...

Modified: "... **our model indicates** that condensates can flow towards distant sites, driven by RNA gradients, and subsequently ..." [line 80]

Previous: Finally, we show that relative clustering, activity, ...

Modified: "Finally, we show **through simulations** that relative clustering, activity, ..." [line 82]

Previous: Spatial clustering of genes is sufficient to recapitulate this phenomenon, irrespective of the coarse-grained representation of cluster shape

Modified: Spatial clustering of genes is sufficient to recapitulate this phenomenon **in our model**, irrespective of the coarse-grained representation of cluster shape [line 143]

Previous: Overall, our results suggest that strong compartmentation and high transcriptional activity gives rise to non-equilibrium morphologies like vacuoles and aspherical droplets

Modified: Overall, our **simulation** results suggest that strong compartmentation and high transcriptional activity gives rise to non-equilibrium morphologies like vacuoles and aspherical droplets [line 233]

Previous: Overall, these results show that activity and compartment strength directly influence condensate size and dynamics

Modified: Overall, these results **from our model indicate** that activity and compartment strength directly influence condensate size and dynamics [line 364]

Figure 1:

- not immediately clear what the middle columns (protein conc., RNA conc., Gene activity) means and what the lines/shades are supposed to say (is the outer black line the nuclear membrane or the condensate boundary?)

We thank the reviewer for their comment. To address it, we have made modifications to Figure 1 in order to improve clarity. We have also added additional description in the caption of the figure clearly stating that the figures represent concentration profiles, etc. The outer black line represents the boundary of a region in space where the concentration of the species is appreciable.

- k_p depends directly on space, so genome organization is not mobile, correct?

Yes, that is correct. We have clarified this in Figure 1B.

- the equation part of the figure is copied over from reference 27 (fig. 4c), also by the authors, this needs to be indicated!

We thank the reviewer for pointing this out and have cited the reference in the figure caption.

Figure 2:

The in-figure legend of 2a seems wrong (Gene compartment and Uniform cannot be found in panel)

We thank the reviewer for having pointed this out. We have changed the in-figure legend.

2c: the overall changes are pretty modest, this would presumably change if system size was larger and larger sigma could be achieved?

Yes. In addition to this, the change in condensate radius between small and large sigma is also going to depend on the initial concentration of the protein/amount of protein in the system.

2f: why is the boundary so wiggly? Is this a numerical artefact? Boundary conditions?

It reflects that the finite sampling of a discrete set of k_r and sigma, which we now clearly state in our caption of Figure 2F.

Figure 4:

2b/c: plotting L_{dr}/r vs peak velocity would be more useful to see the transition from no to flow regime. Also, is this an actual jump or a continuous increase in flow velocity? If the latter, then the threshold needs to be discussed, since it is somewhat arbitrary.

We thank the reviewer for their comment and have addressed their question by significantly revising our presentation of the related text and figures. Following the reviewer's suggestion, we first plotted L_{dr}/r versus peak velocity as we change distance and mobility as shown below and observe the change from flow to no-flow.:

Reviewer figure 4. (a) Peak flow velocity vs. the ratio of reaction-diffusion length scale and the distance between site of gene activity and condensate (L_{dr}/r). L_{dr} is calculated as $(M/\langle k_T \rangle)^{0.5}$ where $\langle k_T \rangle$ is the average RNA transcription rate in the domain. This plot is generated by varying the RNA mobility M_r . The condensates start to flow when L_{dr}/r are or order 1. **(b)** Peak flow velocity vs. the ratio of reaction-diffusion length scale and the distance between site of gene activity and condensate (L_{dr}/r). L_{dr} is calculated as $(M/\langle k_T \rangle)^{0.5}$ where $\langle k_T \rangle$ is the average RNA transcription rate in the domain. This plot is generated by varying the distance between the site of gene activity and condensate (r). The condensates start to flow when L_{dr}/r are or order 1.

We find that the increase in flow velocity is a continuous change but happens rather quickly and thus appears abrupt. To better understand the origin of this phenomenon, we developed a new theoretical analysis method to better approximate the RNA gradient (SI Theory of flow). We derive an analytical estimate of the RNA gradient, including an improved estimate of L_{dr} that reflects the complex dependencies on RNA mobility, synthesis, and degradation. This nuanced estimate of the RNA gradient better captures the continuous change in flow velocity (Figure 4) as well as the absence of flow (or negligible motion) at large r . Further, our model predicts how maximum flow velocities change with varying RNA mobility, distance, and RNA degradation rates (Figure 4C-E). With this improved understanding of condensate flow, we have removed the less nuanced understanding of L_{dr}/r as shown in the above reviewer figure from the main figures. If the reviewer still feels this approximate estimate is helpful, we are happy to include it as a supplementary figure.

Is flow coupled to a significant increase in condensate volume? This would be a potential experimental check for this mechanism.

We thank the reviewer for their comment. The size of the condensate at steady state is set by the parameters k_T and σ of the gene locus. If we start with a nucleated droplet of a larger size than this, the condensate will shrink. If we start with a droplet that is smaller than this size, it will expand. Therefore, we would like to be careful and not make any statements about size changes during directed flow. We have changed the caption of figure 4A to avoid this confusion.

Generally, the supplement fig. S3 seems more informative than main fig. 4, but would probably need a finer grained simulation.

We thank the reviewer for their comment. We have now put the flow velocity plots in the main figures (Figure 4) and have run additional simulations to produce the plot at a higher resolution (S5A-B).

REVIEWERS' COMMENTS

Reviewer #2 (Remarks to the Author):

The authors have largely addressed my main comments. Specifically, they have expanded the description of the model in the main text and explained the rationale behind the choice of the constraints of most model parameters. Additional model parameters have been specified in the Supplementary Information and several references have been added. The simulated flow velocity (Fig. 4) has been compared with one experimental value and a paragraph about model limitations has been added. This reviewer still believes that a more extensive comparison with experimental data would have strengthened the work, but overall this set of model predictions represents an interesting framework to guide and possibly rationalize future experiments.

Minor

- Table S1: $Da \gg 1$: it is rather vague, this can be any number
- Caption Fig. 4B: "Compare this to figure 2D in Kim et al". This is unclear. I would suggest to make the comparison explicit here, as in the main text.
- Figure S3d: why nucleation time does not increase monotonically in the range of kr between 100 and 500?
- I leave this decision to the authors, but I would suggest adding in the title something along the line "Modelling organization and regulation of..."

Summary of reviewer response

We thank the reviewers for their response and positive assessment of our revised study. We have addressed the remaining minor concerns of Reviewer 2 and outline them point-by-point below.

Reviewer #2 (Remarks to the Author):

The authors have largely addressed my main comments. Specifically, they have expanded the description of the model in the main text and explained the rationale behind the choice of the constraints of most model parameters. Additional model parameters have been specified in the Supplementary Information and several references have been added. The simulated flow velocity (Fig. 4) has been compared with one experimental value and a paragraph about model limitations has been added. This reviewer still believes that a more extensive comparison with experimental data would have strengthened the work, but overall this set of model predictions represents an interesting framework to guide and possibly rationalize future experiments.

Minor:

- Table S1: $Da \gg 1$: it is rather vague, this can be any number.

We now specify the ranges we simulated explicitly in Table S1.

- Caption Fig. 4B: "Compare this to figure 2D in Kim et al". This is unclear. I would suggest to make the comparison explicit here, as in the main text.

The comparison is now made explicit as it was in the main text.

- Figure S3d: why nucleation time does not increase monotonically in the range of k_r between 100 and 500?

We thank the reviewer for making this observation. When we looked at the condensate radius on the accompanying left panel in S3D, one clearly notes a monotonically decreasing size with gene activity. This discrepancy led us to double check the original panel and we identified that the nucleation time panel (on the right) was incorrectly plotted by accident. We corrected this (including through validation by re-running simulations again) error - as shown in the updated version of the current Figure S3SD, the nucleation time monotonically decreases with increasing transcription activity as expected from the condensate size. This finding is intuitive, as the rate of transcription increases, the RNA concentration required to nucleate a dense phase is achieved quicker and thus nucleation times decrease with increasing activity.

- I leave this decision to the authors, but I would suggest adding in the title something along the line "Modelling organization and regulation of..."

We propose the following title that we believe addresses the reviewer's concern: "A model for organization and regulation of nuclear condensates by gene activity"